# Trimmed Maximum Likelihood Estimation for Robust Learning in Generalized Linear Models

**Pranjal Awasthi**
Google Research
pranjalawasthi@google.com

**Abhimanyu Das**
Google Research
abhidas@google.com

**Weihao Kong**
Google Research
weihaokong@google.com

**Rajat Sen**
Google Research
senrajat@google.com

## Abstract

We study the problem of learning generalized linear models under adversarial corruptions. We analyze a classical heuristic called the *iterative trimmed maximum likelihood estimator* which is known to be effective against *label corruptions* in practice. Under label corruptions, we prove that this simple estimator achieves minimax near-optimal risk on a wide range of generalized linear models, including Gaussian regression, Poisson regression and Binomial regression. Finally, we extend the estimator to the more challenging setting of *label and covariate corruptions* and demonstrate its robustness and optimality in that setting as well.

## 1 Introduction

Generalized linear models (GLMs) are an elegant framework for statistical modeling of data and are widely used in many applications [NW72, DB18, MN19]. A generalized linear model captures the relationship of labels (or observations) $y$ to covariates $\mathbf{x}$ via the conditional density $f(y|\beta^\top \mathbf{x}) \propto \exp(y \cdot \beta^\top \mathbf{x} - b(\beta^\top \mathbf{x}))$. Here $\beta$ is the parameter of the model. Parameter estimation in GLMs is done via the standard maximum likelihood estimation (MLE) paradigm and has been extensively studied [NW72]. While GLMs offer a mathematically tractable formulation for statistical modeling, real data rarely satisfies the generative process of a GLM and as a result there has been considerable interest in developing robust learning algorithms for GLMs [LW11, NRWY12, PSBR18]. The predominant way to model misspecification or adversarial corruptions in the data is Huber's $\epsilon$-contamination model [Hub11] and its recent extensions that allow for stronger adversaries [DKK+19]. These models assume that a small $\epsilon$ fraction of the data is corrupted by an adversary. Furthermore, the adversary can be restricted to either only corrupting the labels $y$, or be allowed to corrupt both the covariates $\mathbf{x}$ and the labels $y$ for an $\epsilon$ fraction of the data.

Various algorithms have been proposed for robust estimation in generalized linear models. One line of work proposed computationally intractable algorithms that are based on non-convex M estimators [Hub11, LW11] or by running tournaments over an exponentially large search space [Yat85]. Another line of work proposes polynomial time algorithms that either achieve sub-optimal error rates [PSBR18] or only apply to restricted settings such as the noise being heavy tailed [ZZ21]. In practical settings a simple heuristic namely the *iterative trimmed estimator* has been shown to work well under settings where only the labels are corrupted [SS19b]. However, from a theoretical perspective the iterative trimmed MLE estimator has only been analyzed under restrictive settings such as when the underlying GLM is a Gaussian regression model.

36th Conference on Neural Information Processing Systems (NeurIPS 2022).

Our key theoretical contribution in this work is a general analysis of the trimmed MLE estimator. In particular, we show that for a broad family of GLMs, and under adversarial corruptions of only the labels, not only does the iterative trimmed MLE estimator enjoy theoretical guarantees, it in fact nearly achieves the minimax error rate! Next, we also consider the more challenging setting of corruptions to both covariates and labels. In the setting where the covariance of the covariate is known, we leverage the same approach in [PJL20] by running the filtering algorithm [DHL19] as a prepossessing step to trim away the abnormal covariates before applying the iterative trimmed MLE estimator. This can simultaneously handle both covariate and label corruptions and nearly achieve minimax error rates. Below we state our main results.

**Theorem 1.1** (Informal Theorem). *Let $S_\epsilon = \{(\mathbf{x}_1, y_1), \ldots, (\mathbf{x}_n, y_n)\}$ be independent and identically distributed samples generated by a generalized linear model with sub-Gaussian $x_i$. Let an $\epsilon$ fraction of the labels be adversarially corrupted. Then, with high probability, the iterative trimmed MLE estimator when given as input $S_\epsilon$ provides the following guarantees:*

- *$O(\sigma\epsilon \log(1/\epsilon))$ parameter estimation error for the Gaussian regression model where $\sigma^2$ is the variance of Gaussian noise on $y$.*

- *$O(\epsilon \exp(\sqrt{\log(1/\epsilon)}))$ parameter estimation error for Poisson regression model.*

- *$O(\frac{1}{\sqrt{m}}\epsilon\sqrt{\log(m/\varepsilon)\log(1/\varepsilon)})$ parameter estimation error for the Binomial regression model with $m$ trials.*

- *$O(\epsilon \log(1/\epsilon))$ parameter estimation error for a general class of smooth and continuous GLMs (includes the Gaussian regression model).*

Previous work [SS19b] analyze the iterative trimmed maximum likelihood estimator under the Gaussian regression settings (least square) where only the labels are corrupted, and proved an $O(\sigma)$ $\ell_2$-error bound for parameter estimation. Other iterative trimming approach such as [BJK15] also achieves only achieve $O(\sigma)$ error in this setting. Our error bound of $O(\sigma\varepsilon \log(1/\varepsilon))$ significantly improved the dependency on the corruption level $\varepsilon$ and nearly matched the minimax lower bound of $\sigma\varepsilon$ [Gao20]. In all the generalized linear models studied in this work, we show that the iterative trimmed maximum likelihood estimator achieves $O(\varepsilon^{1-\delta})$ error for any $\delta > 0$, which matches the minimax lower bound $\Omega(\varepsilon)$ up to a sub-polynomial factor.

Next, we present our second main result that can simultaneously handle both covariate and label corruptions and nearly achieve minimax error rate.

**Theorem 1.2** (Informal Theorem). *Let $S_\epsilon = \{(\mathbf{x}_1, y_1), \ldots, (\mathbf{x}_n, y_n)\}$ be independent and identically distributed samples generated by a generalized linear model with sub-Gaussian $x_i$ whose covariance is known. Let an $\epsilon$ fraction of the labels and covariates be adversarially corrupted. After a preprocessing step, the iterative trimmed MLE with high probability achieves the same parameter estimation recovery bounds as in Theorem 1.1 above.*

Our algorithm requires the covariance matrix of the covariate distribution being identity or known. Thus the error rate we get is incomparable to the results in the general covariance settings.

**Outline of the paper.** In Section 2 we discuss related work. We define preliminaries in Section 3 followed by the iterative trimmed MLE algorithm, formal results, and proof sketches for the label corruption case in Section 4. In Section 5, we introduce our algorithm, results and proof sketches for the sample corruption case. We defer proofs to the Appendix.

## 2 Related Work

There is a vast amount of literature in statistics, machine learning and theoretical computer science on algorithms that are robust to adversarial corruptions and outliers. Classical works in the Huber's contamination model present algorithms for general robust estimation that obtain near optimal error rates [Hub11, Yat85]. Minimax optimal but computationally inefficient robust estimators have been established in the works of [Tuk75, Yat85, CGR15, Gao20] for a variety of problems such as mean and covariance estimation. In recent years there has also been a line of work in designing computationally efficient algorithms for handling adversarial corruptions [LRV16, DKK+19, CSV17, BJK15, CKMY22, CAT+20, DKSS21].

Several special cases of generalized linear model has been studied extensively from the robustness perspective. There is a long line of work on designing robust algorithms for the linear least squares problem. This corresponds to the special case of the GLM being a Gaussian regression model [SBRJ19, BJK15, BJKK17, KKM18, CCM13, BP21, PJL20, DKS19]. When the covariate follows from a sub-Gaussian distribution, previous best result [PJL20] achieves $\ell_2$ error $\sigma\varepsilon\sqrt{\log(1/\varepsilon)}$ using Huber regression while we show iterative thresholding achieves $\sigma\varepsilon\log(1/\varepsilon)$ error. Another special case of GLMs that has been studied from a robustness perspective is the logistic regression model [FXMY14, PSBR18, CKMY20]. Our analysis of the iterative trimmed MLE estimator on binomial regression matches the best known error guarantees for this setting.

The work of [PSBR18] proposes a general procedure for robust gradient descent and shows that one can use this to design robust estimation algorithms for a abroad class of GLMs. Building upon these works, the authors in [JLST21] present a nearly linear time algorithm for GLMs. Compared to our work, [PSBR18] assumes $x$ has bounded 8th moment, [JLST21] assumes $x$ is 2-4 hypercontractive, and both papers achieve an $O(\sqrt{\epsilon})$ error guarantee with $\varepsilon$-fraction of corruptions while our algorithm achieves a better $O(\varepsilon)$ guarantee under the stronger sub-Gaussian assumptions. In addition, for GLMs, both papers assume a uniform upper bound (and lower bound) on the second order derivative of function $b(\cdot)$, which is not satisfied by the widely used Poisson and Binomial regression studied in this paper. In a similar setting, [ZZ21] proposed a reweighted MLE estimator for dealing with settings where the covariates are heavy tailed but not corrupted by an adversary.

Iterative thresholding is a longstanding heuristic for robust linear regression that dates back to Legendre [LS59]. It's theoretical property in the non-asymptotic regime is first studied in [BJK15], which shows a $O(\sigma)$ error bound for sufficiently small label corruption level when the label noise is $N(0, \sigma^2)$. The iterative thresholding algorithm is later extended and analyzed in the *oblivious* label corruption setting [BJKK17, SBRJ19], and is shown to provide consistent estimate even when the corruption level goes to $1$. [PJL20] extended the iterative thresholding algorithm to the heavy-tailed covariate setting and can simultaneously handle both labels and covariate corruptions. In addition, [PJL20] adapted the implicit result in [BJKK17] to show that iterative thresholding algorithm achieves $O(\sigma\sqrt{\varepsilon})$ error rate for sub-Gaussian covariate. [SS19b] empirically demonstrated the effectiveness of the trimmed loss estimator under labels corruptions, and also proved theoretical guarantees in a special class of GLMs, which, however, also has $O(\sigma)$ error when specialized to the Gaussian noise setting. [SS19a] studied the trimmed loss estimator in the mixed linear regression setting. Finally, [CKMY22] proposed an alternating minimization algorithm for the fixed design linear regression setting with Huber contamination on the labels, which is different from our strong contamination model (with adaptive replacement) for generalized linear model. Their algorithm incorporates a semidefinite programming in the set selection step, and achieves a near-optimal $\tilde{O}(\sigma\varepsilon)$ error. On a very high level, our proof follows from a similar framework as in [CKMY22].

[PJL20] first proposed a generic approach to modify an estimator which is robust against label corruption into one that is robust against simultaneous label and covariate corruptions by running a covariate filter algorithm [DKP20, DK19] as a preprocessing step. In the Gaussian regression setting, our Algorithm 2 is identical to Algorithm 3 of [PJL20] and the difference is in the improved error rate. In particular, Lemma 4.1 in [PJL20] implies a $O(\sigma\sqrt{\varepsilon})$ error bound while we proved a $O(\sigma\varepsilon\log(1/\varepsilon))$ error bound.

# 3 Preliminaries

In this section, we formally introduce the robust generalized linear model studied in this paper. First we define the classical generalized linear model as follows.

**Definition 3.1** (Generalized linear model). *We say that $(\mathbf{x}, y)$ follows from a generalized linear model if there exist function $b(\cdot)$, and function $c(\cdot)$ such that the probability density function of $y$ equals*

$$f(y|\beta^\top \mathbf{x}) = c(y)\exp(y \cdot \beta^\top \mathbf{x} - b(\beta^\top \mathbf{x})).$$

*The derivative $b'(\cdot)$ is called mean function as $\mathbb{E}[y|\beta^\top \mathbf{x}] = b'(\beta^\top \mathbf{x})$. The second order derivative $b''(\cdot)$ is called variance function as $\mathrm{Var}[y|\beta^\top \mathbf{x}] = b''(\beta^\top \mathbf{x})$*

Here we provide three commonly used examples of generalized linear models studied in this paper.

**Definition 3.2.** *Commonly used examples of generalized linear model*

- **Gaussian regression**: $b(\theta) = \frac{1}{2}\theta^2$, $c(y) = \frac{\exp(-y^2/2)}{\sqrt{2\pi}}$

- **Poisson regression**: $b(\theta) = \exp(\theta)$, $c(y) = \frac{1}{y!}$

- **Binomial regression**: *Let $m$ be the number of trials of the binomial distribution.* $b(\theta) = m\log(1 + \exp(\theta))$, $c(y) = \binom{m}{y}$

We consider the sub-Gaussian random design setting in this work, i.e., each covariate $\mathbf{x}_i$ is drawn i.i.d. from a sub-Gaussian distribution with zero mean and covariance $\Sigma$.

**Definition 3.3** (Sub-Gaussian design). *We assume that each covariate $\mathbf{x}_i$ is drawn independently from a zero mean, covariance $\Sigma$ sub-Gaussian distribution with sub-Gaussian norm $1$, namely, $\mathbb{E}[\mathbf{x}_i] = 0$, $\mathbb{E}[\mathbf{x}_i\mathbf{x}_i^\top] = \Sigma$ and for all $\mathbf{v} \in \mathbb{R}^d$,*

$$\Pr(\mathbf{v}^\top\mathbf{x}_i \geq t) \leq \exp(-t^2).$$

Having introduced the generation model of the good data, now we describe the corruption model where the adversary is allowed to corrupt a small fraction of the data points.

**Definition 3.4** (Corruption model). *We consider two different data generation model with $\varepsilon$ fraction of adversarial corruptions:*

- **Label corruption model**: *Given $n$ i.i.d. sample $\{(\mathbf{x}_i, y_i)\}_{i=1}^n$ generated by a generalized linear model. The adversary is allowed to inspect the sample, and replace a total of $\varepsilon \cdot n$ labels $y_i$ with arbitrary values.*

- **Sample corruption model**: *Given $n$ i.i.d. sample $\{(\mathbf{x}_i, y_i)\}_{i=1}^n$ generated by a generalized linear model. The adversary is allowed to inspect the sample, and replace a total of $\varepsilon \cdot n$ data points $(\mathbf{x}_i, y_i)$ with arbitrary values.*

*We call the corrupted dataset $S = T \cup E$ where $T$ contains the set of remaining uncorrupted data points, and $E$ contains the set of data points that is controlled by the adversary.*

The goal of our algorithm is recovering the underlying regression coefficient $\beta^*$ from a set of examples with $\varepsilon$ fraction corrupted under $\ell_2$ error metric, i.e., $\|\hat{\beta} - \beta^*\|$. We assume $\|\beta^*\| \leq R$ for a constant $R$, and $\varepsilon <= c$ for a sufficiently small constant $c$. For simplicity of the presentation, throughout this paper, we assume $\Sigma = I_d$. We remark that our algorithm for *label corruption model* applies to the general covariance setting and is able to achieve small estimation error in terms of $\|\Sigma^{1/2}(\hat{\beta} - \beta^*)\|_2$ since we can always (implicitly) whiten the data to apply our analysis (see Section C for more details). On the other hand, the algorithm for *sample corruption model* only works with the knowledge of $\Sigma$.

## 4    Label Corruption

In this section, we formally describe our algorithm and proof-sketch for the label corruption setting.

### 4.1    Algorithm

We start with defining the trimmed maximum likelihood estimator, which is a simple and natural heuristic for robustly learning generalized linear model.

**Definition 4.1** (Trimmed maximum likelihood estimator). *Given a set of data points $S = \{(\mathbf{x}_i, y_i)\}_{i=1}^n$, define the trimmed maximum likelihood estimator as*

$$\hat{\beta}(S) = \min_\beta \min_{\hat{S} \subset S, |\hat{S}| = (1-\varepsilon)n} \sum_{(\mathbf{x}_i, y_i) \in \hat{S}} -\log f(y_i | \beta^\top\mathbf{x}_i)$$

In the setting of generalized linear model, the objective of trimmed maximum likelihood estimator is a biconvex problem in $\hat{S}$ and $\beta$ but not jointly convex. The following alternating minimization algorithm is a simple heuristic to approximate the trimmed maximum likelihood estimator whose similar form has been studied in [BJK15, BJKK17, SS19b, CKMY22].

---

**Algorithm 1:** Alternating minimization of trimmed maximum likelihood estimator

---
**Input:** Set of examples $S = \{(\mathbf{x}_1, y_1), \ldots, (\mathbf{x}_n, y_n)\}, \varepsilon, \eta, R$
**Output:** $\hat{\beta}$
**1** $S^{(0)} \leftarrow \arg\min_{T \subset [n]:|T|=(1-\varepsilon)n} \sum_{i \in T} |y_i|$;
**2** $\hat{\beta}^{(1)} \leftarrow 0$;
**3 for** $t = 1$ **to** $\infty$ *do* **do**
**4** $\quad$ Choose $\hat{S}^{(t)} = \arg\min_{T \subset S^{(0)}:|T|=(1-2\varepsilon)n} \sum_{i \in T} -\log f(y_i|\langle \hat{\beta}^{(t)}, \mathbf{x}_i \rangle)$;
**5** $\quad$ Compute $\hat{\beta}^{(t+1)} = \arg\min_{\beta, \|\beta\| \leq R} \sum_{i \in \hat{S}^{(t)}} -\log f(y_i|\langle \beta, \mathbf{x}_i \rangle)$;
**6** $\quad$ **if** $\frac{1}{n} \sum_{i \in \hat{S}^{(t)}} -\log f(y_i|\langle \hat{\beta}^{(t+1)}, \mathbf{x}_i \rangle) > \frac{1}{n} \sum_{i \in \hat{S}^{(t)}} -\log f(y_i|\langle \hat{\beta}^{(t)}, \mathbf{x}_i \rangle) - \eta$ **then**
**7** $\quad\quad \lfloor$ Return $\beta^{(t)}$

---

The algorithm starts by naively pruning out $\varepsilon \cdot n$ data points whose labels have the largest magnitude. Then each round of the alternating minimization algorithm has two steps. In optimizing over set $S$, we find the set $\hat{S}^{(t)}$ of size $(1-\varepsilon)n$ with the best likelihood. In optimizing over $\beta$, we find regression coefficient $\beta^{(t)}$ which maximizes the likelihood on the current set of data $\hat{S}^{(t)}$. The algorithm terminates and outputs $\beta$ when the likelihood no longer improves by more than $\eta$. It is clear that the algorithm terminates in $O(1/\eta)$ rounds when the log-likelihood is bounded. Since the original trimmed maximum likelihood estimator is a biconvex optimization problem in $S, \beta$ which is not jointly convex, our algorithm does not guarantee to return a global optimal solution. Nonetheless, as we showed, it does return a first order stationary point which will be close to the true coefficient $\beta^*$. It is worth noting that some recent papers on robust statistics [CDK+21, CDGS20, ZJS22] show similar nice statistical properties of an approximate first order stationary point for non-convex optimization problems.

We present the guarantee of our algorithm for Gaussian, Poisson, Binomial regression, and a broad class of generalized linear models.

**Theorem 4.2** (Gaussian regression with label corruption). *Let $S = \{\mathbf{x}_i, y_i\}_{i=1}^n$ be a set of data points generated by a Gaussian regression model with $y_i = \langle \mathbf{x}_i, \beta^* \rangle + \eta_i, \eta_i \sim N(0, \sigma^2)$, sub-Gaussian design, with $\varepsilon_c$-fraction of label corruption and $n = \Omega(\frac{d+\log(1/\delta)}{\varepsilon^2})$. With probability $1 - \delta$, Algorithm 1 with parameters $\varepsilon = \varepsilon_c, \eta = \varepsilon_c^2, R = \infty$ terminate within $O(\frac{1}{\min(1,\sigma^2)\varepsilon_c^2})$ iterations, and output an estimate $\hat{\beta}$ such that*

$$\|\hat{\beta} - \beta^*\| = O(\sigma \varepsilon_c \log(1/\varepsilon_c))$$

**Theorem 4.3** (Poisson regression with label corruption). *Let $S = \{\mathbf{x}_i, y_i\}_{i=1}^n$ be a set of data points generated by a Poisson regression model with sub-Gaussian design, with $\varepsilon_c$-fraction of label corruption and $n = \Omega(\frac{d}{\varepsilon^2})$. With probability $0.99$, Algorithm 1 with parameters $\varepsilon = 2\varepsilon_c, \eta = \varepsilon_c^2/(dn)$, contant $R \geq \|\beta^*\|$ terminate within $dn/\varepsilon_c^2$ iterations, and output an estimate $\hat{\beta}$ such that*

$$\|\hat{\beta} - \beta^*\| = O(\varepsilon_c \exp(\sqrt{\log(1/\varepsilon_c)}))$$

**Theorem 4.4** (Binomial regression with label corruption). *Let $S = \{\mathbf{x}_i, y_i\}_{i=1}^n$ be generated by a Binomial regression model with sub-Gaussian Design, with $\varepsilon_c$-fraction of label corruption and $n = \Omega(\frac{d+\log(1/\delta)}{\varepsilon^2})$. With probability $1 - \delta$, Algorithm 1 with parameters $\varepsilon = \varepsilon_c, \eta = \varepsilon_c^2/m$, constant $R \geq \|\beta^*\|$ terminate within $m^2/\varepsilon_c^2$ iterations, and output an estimate $\hat{\beta}$ such that*

$$\|\hat{\beta} - \beta^*\| = O\left(\varepsilon_c \sqrt{\frac{\log(m/\varepsilon_c)\log(1/\varepsilon_c)}{m}}\right)$$

**Theorem 4.5** (A class of generalized linear model with label corruption). *Let $S = \{\mathbf{x}_i, y_i\}_{i=1}^n$ be generated by a generalized linear model with sub-Gaussian Design, with $\varepsilon_c$-fraction of label corruption and $n = \Omega(\frac{d+\log(1/\delta)}{\varepsilon^2})$. Assuming that $C_0 \leq b''(\cdot) \leq C$ for non-zero constants $C_0, C$, $b(0) = 0, b'(0) = 0$, and $\log(c(y)) = O(\log(1/\varepsilon_c)), \forall y \leq \Theta(\sqrt{\log(1/\varepsilon_c)})$, then with probability*

$1-\delta$, *Algorithm 1 with parameters* $\varepsilon = \varepsilon_c, \eta = \varepsilon_c^2, R = \infty$ *terminates within* $\log(1/\varepsilon_c)/\varepsilon_c^2$ *iterations, and output an estimate* $\hat{\beta}$ *such that*

$$\|\hat{\beta} - \beta^*\| = O(\varepsilon_c \log(1/\varepsilon_c))$$

## 4.2 Proof Sketch

We provide an intuitive proof sketch for the above theorems (the full proofs are deferred to Section A in the Appendix). The high level proof framework is similar to [CKMY22], although the details are drastically different since the focus of our paper is on random design with strong contamination for a wide range of generalized linear model while [CKMY22] focuses on linear (least square) regression with Huber corruption on the labels.

The guarantee of our alternating minimization algorithm relies on two claims: First, the algorithm returns an approximate stationary point $\hat{\beta}$. Second, any approximate stationary point will be close to the true coefficient $\beta^*$. In this section, we will present high level intuition of the proof of the two claims.

**Alternating minimization algorithm returns an approximate stationary point.** First we define the first order approximate stationary point as follows. Let $\hat{\beta} \in \mathbb{R}^d$ be a regression coefficient vector and $\hat{S}$ contains the set of datapoints of size $(1 - \varepsilon)n$ with the largest log-likelihood under $\hat{\beta}$. We call $\hat{\beta}$ a $\gamma$-approximate stationary point if

$$\frac{1}{n} \sum_{i \in \hat{S}} \nabla_\beta \log f(y_i | \langle \hat{\beta}, \mathbf{x}_i \rangle)^\top \frac{(\beta^* - \hat{\beta})}{\|\beta^* - \hat{\beta}\|} \leq \gamma$$

i.e., the gradient of the log-likelihood projected along the $\beta^* - \hat{\beta}$ direction is small. Our goal is to show when the algorithm terminates, that is when $\hat{\beta}$ can not be improved by more than $\eta$, the gradient along the $\beta^* - \hat{\beta}$ must be small. This is clear where the empirical log-likelihood function is smooth, simply because if the gradient is large, one can improve the log-likelihood by more than $\eta$ which will result in a contradiction. The smoothness (norm of the Hessian matrix) of the empirical log-likelihood in generalized linear model is directly related to the range of $b''(\theta)$. In particular, $b''(\theta)$ is bounded for Gaussian regression and Binomial regression.

However, a problem arises for Poisson regression where $b''(\theta) = \exp(\theta)$ becomes extremely large for large $\theta$, which results in non-smooth curvature for empirical log-likelihood. We overcome this difficulty by leveraging the special property of function $b(\theta) = \exp(\theta)$ in the Poisson regression setting. Observing that the derivative $b'(\theta)$ is equal to second order derivative $b''(\theta)$ for Poisson regression, the gradient along the $\beta^* - \hat{\beta}$ direction can not be small when the second order derivative along the $\beta^* - \hat{\beta}$ direction gets large, which will result in a more than $\eta$ improvement of the log-likelihood by moving toward $\beta^*$ and therefore a contradiction. Hence, the second order derivative along the $\beta^* - \hat{\beta}$ direction must be small, and we blue have the same argument as in the smooth objective function setting.

**Any approximate stationary point will be close to the true coefficient.** Let us first write down the $\gamma$-approximate stationary condition for generalized linear model as

$$\frac{1}{n} \sum_{i \in \hat{S}} \nabla_\beta \log f(y_i | \langle \hat{\beta}, \mathbf{x}_i \rangle)^\top (\beta^* - \hat{\beta}) = \frac{1}{n} \sum_{i \in \hat{S}} (y_i - b'(\hat{\beta}^\top \mathbf{x}_i))(\beta^* - \hat{\beta})^\top \mathbf{x}_i \leq \gamma \|\beta^* - \hat{\beta}\|$$

Recall that $T$ contains the set of uncorrupted data points, and $E$ contains the set of data points that is controlled by the adversary. Split $\hat{S}$ into $\hat{S} \cap T$ and $\hat{S} \cap E$, and rearrange the terms we get

$$\frac{1}{n} \sum_{i \in \hat{S} \cap T} (y_i - b'(\hat{\beta}^\top \mathbf{x}_i))(\beta^* - \hat{\beta})^\top \mathbf{x}_i \leq -\frac{1}{n} \sum_{i \in \hat{S} \cap E} (y_i - b'(\hat{\beta}^\top \mathbf{x}_i))(\beta^* - \hat{\beta})^\top \mathbf{x}_i + \gamma \|\beta^* - \hat{\beta}\|.$$

To obtain an upper bound on $\|\beta^* - \hat{\beta}\|$, we will prove a lower bound in terms of $\|\beta^* - \hat{\beta}\|$ on the left hand side, and an upper bound in terms of $\|\beta^* - \hat{\beta}\|$ on the right hand side. Finally we will combine the upper and lower bound into an upper bound on $\|\beta^* - \hat{\beta}\|$.

**Lower bound on the LHS.** Note that $\hat{S} \cap T$ contains uncorrupted data points. The high level intuition is that since mean of $y_i$ is $b'(\beta^{*\top}\mathbf{x}_i)$ and $b'(\cdot)$ is monotone, $\left(y_i - b'(\beta^{*\top}\mathbf{x}_i)\right)(\beta^* - \hat{\beta})^\top\mathbf{x}_i$ should be roughly $O\left(\left((\beta^* - \hat{\beta})^\top\mathbf{x}_i\right)^2\right)$, and $\left((\beta^* - \hat{\beta})^\top\mathbf{x}_i\right)^2$ should be proportional to $\|\beta^* - \hat{\beta}\|^2$ given enough samples. More formally, we will decompose the LHS as

$$\frac{1}{n}\sum_{i \in \hat{S} \cap T}(y_i - b'(\hat{\beta}^\top\mathbf{x}_i))(\beta^* - \hat{\beta})\mathbf{x}_i$$
$$=\frac{1}{n}\sum_{i \in \hat{S} \cap T}\left(y_i - b'(\beta^{*\top}\mathbf{x}_i)\right)(\beta^* - \hat{\beta})^\top\mathbf{x}_i + \frac{1}{n}\sum_{i \in \hat{S} \cap T}\left(b'(\beta^{*\top}\mathbf{x}_i) - b'(\hat{\beta}^\top\mathbf{x}_i)\right)(\beta^* - \hat{\beta})^\top\mathbf{x}_i.$$

The first term contains a $(1 - \varepsilon)$ fraction of uncorrupted random examples sampled from a zero mean distribution with certain tail bound, e.g. sub-exponential for Gaussian and Binomial regression, $k$-th moment bound for Poisson regression.

Therefore, we can apply *resilience* property to bound the first term. Overall, we heavily utilize the *resilience* property of the sample set that is drawn from "nice" distributions. Take sample mean as an example, *resilience* [SCV17, ZJS19] (also known as *stability* [DK19]) dictates that given a large enough sample set $S = \{\mathbf{x}_i\}_{i=1}^n$, the sample mean of any large enough subset of $S$ will be close to each other. We define mean resilience formally here:

**Definition 4.6** (Resilience). *Given a sample set $S = \{\mathbf{x}_i\}_{i=1}^n$, suppose for any $T \subset S, |T| \geq (1-\varepsilon)n$, it holds that $\|\frac{1}{|T|}\sum_{i \in T}\mathbf{x}_i - \frac{1}{|S|}\sum_{i \in S}\mathbf{x}_i\| \leq \tau$, then we call the set $S$ satisifes $(\varepsilon, \tau)$-resilience.*

Specifically, under sub-Gaussian distribution, a set $S$ of i.i.d. samples with size $n = \Omega(d/\varepsilon^2)$ satisfies $(\varepsilon, \varepsilon\sqrt{\log(1/\varepsilon)})$ resilience with high probability. Resilience property applies to sub-exponential and $k$-th moment bounded distribution as well, and this gives us a way to control the behavior of any subset of good data.

For the second term, we prove that $\frac{1}{n}\sum_{i \in \hat{S} \cap T}b(\beta^\top\mathbf{x}_i)$ is a strongly convex function again using the resilience property, which implies $\frac{1}{n}\sum_{i \in \hat{S} \cap T}\left(b'(\beta^{*\top}\mathbf{x}_i) - b'(\hat{\beta}^\top\mathbf{x}_i)\right)(\beta^* - \hat{\beta})^\top\mathbf{x}_i = \Omega(\|\beta^* - \hat{\beta}\|^2)$.

**Upper bound on the RHS.** To upper bound $-\frac{1}{n}\sum_{i \in \hat{S} \cap E}(y_i - b'(\hat{\beta}^\top\mathbf{x}_i))(\beta^* - \hat{\beta})^\top\mathbf{x}_i$, we will prove an upper bound on $\sqrt{\frac{1}{n}\sum_{i \in \hat{S} \cap E}(y_i - b'(\hat{\beta}^\top\mathbf{x}_i))^2}$ and $\sqrt{\frac{1}{n}\sum_{i \in \hat{S} \cap E}((\beta^* - \hat{\beta})^\top\mathbf{x}_i)^2}$ separately, then apply Cauchy-Schwarz inequality. The key difficulty is bounding $\sqrt{\frac{1}{n}\sum_{i \in \hat{S} \cap E}(y_i - b'(\hat{\beta}^\top\mathbf{x}_i))^2}$, as it contains corrupted data points controlled by an adversary, which does not follow any good property possessed by the good stochastic data. However, since $\hat{S}$ contains $(1 - \varepsilon)n$ datapoints with the largest log-likelihood under $\hat{\beta}$, we can argue that

$$\sum_{i \in \hat{S} \cap E} -\log f(y_i|\langle\hat{\beta}, \mathbf{x}_i\rangle) \leq \sum_{i \in T \setminus \hat{S}} -\log f(y_i|\langle\hat{\beta}, \mathbf{x}_i\rangle)$$

or even

$$\max_{i \in \hat{S} \cap E} -\log f(y_i|\langle\hat{\beta}, \mathbf{x}_i\rangle) \leq \min_{i \in T \setminus \hat{S}} -\log f(y_i|\langle\hat{\beta}, \mathbf{x}_i\rangle)$$

since otherwise one can replace the data points in $\hat{S} \cap E$ by the ones in $T \setminus \hat{S}$ to form a new set with better likelihood than $\hat{S}$. This gives us an upper bound on the negative log-likelihood of $y_i$ in $\hat{S} \cap E$. Therefore we adopt a two step approach to upper bound the $\sqrt{\frac{1}{n}\sum_{i \in \hat{S} \cap E}((\beta^* - \hat{\beta})^\top\mathbf{x}_i)^2}$. First we prove an upper bound on the negative log-likelihood on $T \setminus \hat{S}$, which becomes a negative log-likelihood bound on $\hat{S} \cap E$ immediately. Second we turn the negative log-likelihood bound into a square error bound.

The two steps vary drastically for different regression models. For the first step of upper bounding the negative log-likelihood, in Gaussian regression we use the resilience property of the quadratic

form of sub-Gaussian random variable. In Poisson regression, we leverage the resilience property of $y_i \mathbf{x}_i$ which is heavy tailed. In Binomial regression, since the distribution only has support size $m$, we directly analyze the resilience of the negative log-likelihood. In general GLMs, due to the generality of the likelihood function, we have to again analyze the resilience of the negative log-likelihood directly. For the second step of turning the log-likelihood bound to a quadratic bound, we get the bound trivially in Gaussian regression since Gaussian likelihood is indeed quadratic. For Poisson and Binomial setting, we have to build a proxy function which lower bound the negative log-likelihood function $-\log f(y_i | \langle \hat{\beta}, \mathbf{x}_i \rangle)$ to connect it to quadratic function. For the general class of GLMs, we leverage the bounds on the $b''(\cdot)$ and $\log(c(y))$ to obtain a quadratic bound.

### 4.3 Proof for Poisson Regression

As an illustrative example, we show how the above proof sketch can be used for Poisson regression to formally prove Theorem 4.3. (Proofs for the other GLMs are deferred to Section A in the Appendix)

**Lemma 4.7** (Approximate stationary point close to $\beta^*$ for Poisson regression)**.** *Given a set of datapoints $S = \{\mathbf{x}_i, y_i\}_{i=1}^n$ generated by a Poisson regression model with $\varepsilon$-fraction of label corruption, and the largest $\varepsilon n$ labels removed. Let $\hat{\beta}$ be a $\max(\varepsilon, \varepsilon^2 / \|\beta^* - \hat{\beta}\|)$-stationary point and $\|\hat{\beta}\| \le R$. Given that $n = \Omega(\frac{d}{\varepsilon^2})$, with probability $0.99$, it holds that*

$$\|\hat{\beta} - \beta^*\| = O(\varepsilon \exp(\Theta(\sqrt{\log(1/\varepsilon)})))$$

Since $b''(\theta) = \exp(\theta)$ is unbounded for Poisson regression, the following lemma (proved in the appendix) shows that alternating minimization algorithm still return an approximate stationary point

**Lemma 4.8** (Algorithm 1 finds an approximate stationary point for Poisson regression)**.** *Given a set of datapoints $S = \{\mathbf{x}_i, y_i\}_{i=1}^n$ generated by a Poisson model with $\varepsilon_c$-fraction of corruption. Assuming that $n = \Omega(\frac{d + \log(1/\delta)}{\varepsilon^2})$, then with probability $1 - \delta$, the output of Algorithm 1 with input parameters $\varepsilon = 2\varepsilon_c, R \ge \|\beta^*\|, \eta = \varepsilon^2/(dn)$, is a $\max(\varepsilon, \frac{2\varepsilon^2}{\|\beta^* - \hat{\beta}\|})$-approximate stationary point.*

*Proof of Theorem 4.3.* Lemma 4.8 implies the output of Algorithm 1 is a $\max(\varepsilon, \frac{\varepsilon^2}{\|\beta^* - \hat{\beta}\|})$ approximate stationary point. Lemma 4.7 then implies that $\|\hat{\beta} - \beta^*\| = O(\varepsilon \exp(\sqrt{\log(1/\varepsilon)}))$. To bound the number of iterations, we need an upper bound on the negative log-likelihood on $\beta = 0$, and a uniform lower bound on the negative log-likelihood. The initial negative log-likelihood is upper bounded by $\frac{1}{n} \sum_{i \in \hat{S}^{(1)}} \log(y_i!) + 1 \le O(\mathbb{E}[y_i^2] + 1) = O(1)$ where $S^{(1)}$ contains the smallest $(1 - \varepsilon)n$ labels. Trivially, there is a $0$ lower bound on the negative log-likelihood for Poisson distribution. Therefore, the algorithm will terminate in $dn/\varepsilon_c^2$ iterations. $\qquad \square$

## 5 Result for Sample Corruption Model

The learning problem becomes much harder in the presence of label and covariate corruption, since it is hard to tell whether a data point is corrupted by simply looking at the likelihood of $y_i$. From a technical level, the resilience condition we leveraged on covariate $\mathbf{x}_i$ in set $E$ breaks down when there is covariate corruption. Luckily, we are able to restore the resilience property by first running the filtering algorithm for robust mean estimation [DHL19]. Specifically, if the covariate distribution has identity (or known) covariance and sub-Gaussian tail, one can apply the filtering algorithm to the $\varepsilon$ corrupted data set, and the resulting data set $\{w_i \mathbf{x}_i\}_{i=1}^n$ will have close to identity covariance and the same resilient condition as an uncorrupted data set. This prepossessing step only takes nearly linear time. This approach is firstly proposed in [PJL20] as a general method to make an algorithm robust against covariate-corruptions.

### 5.1 Algorithm

The guarantee of Algorithm 2 is formalized in the following theorems.

**Theorem 5.1** (Gaussian regression with sample corruption)**.** *Given a set of datapoints $S = \{\mathbf{x}_i, y_i\}_{i=1}^n$ generated by a Gaussian regression model with $y_i = \langle \mathbf{x}_i, \beta^* \rangle + \eta_i, \eta_i \sim N(0, \sigma^2)$, sub-Gaussian design, $\varepsilon_c$-fraction of sample corruption and $n = \Omega(\frac{d + \log(1/\delta)}{\varepsilon^2})$. With probability*

---

**Algorithm 2:** Alternating minimization of trimmed maximum likelihood estimator in sample corruption model

---

**Input:** Set of examples $S = \{(\mathbf{x}_1, y_1), \ldots, (\mathbf{x}_n, y_n)\}, \Sigma, \varepsilon, \eta, R$

**Output:** $\hat{\beta}$

**1** $S_0 \leftarrow \{(\Sigma^{-1/2}\mathbf{x}_1, y_1), \ldots, (\Sigma^{-1/2}\mathbf{x}_n, y_n)\}$ // Whiten the covariates.

**2** $S' \leftarrow \text{Filtering}(S, \varepsilon)$ // Algorithm 4 in [DHL19]

**3** $\hat{\beta} \leftarrow \text{Algorithm } 1(S', \varepsilon, \eta, R)$;

**4** Return $\hat{\beta}$;

---

$1 - \delta$, *Algorithm 2 with parameters* $\varepsilon = \varepsilon_c, \eta = \varepsilon_c^2, R = \infty$ *terminate within* $O(\frac{1}{\min(1,\sigma^2)\varepsilon_c^2})$ *iterations, and output an estimate* $\hat{\beta}$ *such that*

$$\|\hat{\beta} - \beta^*\| = O(\sigma\varepsilon_c \log(1/\varepsilon_c))$$

**Theorem 5.2** (Poisson regression with sample corruption). *Given a set of datapoints* $S = \{\mathbf{x}_i, y_i\}_{i=1}^n$ *generated by a Poisson regression model with* $\varepsilon_c$*-fraction of label corruption and* $n = \Omega(\frac{d}{\varepsilon^2})$. *With probability* 0.99, *Algorithm 2 with parameters* $\varepsilon = 2\varepsilon_c, \eta = \varepsilon_c^2/(dn), R \geq \|\beta^*\|$ *terminate within* $dn/\varepsilon_c^2$ *iterations, and output an estimate* $\hat{\beta}$ *such that*

$$\|\hat{\beta} - \beta^*\| = O(\varepsilon_c \exp(\sqrt{\log(1/\varepsilon_c)}))$$

**Theorem 5.3** (Binomial regression with sample corruption). *Given a set of datapoints* $S = \{\mathbf{x}_i, y_i\}_{i=1}^n$ *generated by a Binomial regression model with* $\varepsilon_c$*-fraction of sample corruption and* $n = \Omega(\frac{d+\log(1/\delta)}{\varepsilon^2})$. *With probability* $1 - \delta$, *Algorithm 2 with parameters* $\varepsilon = \varepsilon_c, \eta = \varepsilon_c^2, R \geq \|\beta^*\|$ *terminate within* $m/\varepsilon_c^2$ *iterations, and output an estimate* $\hat{\beta}$ *such that*

$$\|\hat{\beta} - \beta^*\| = O(\varepsilon_c\sqrt{\frac{\log(m/\varepsilon_c)\log(1/\varepsilon_c)}{m}})$$

**Theorem 5.4** (A class of generalized linear model with sample corruption). *Let* $S = \{\mathbf{x}_i, y_i\}_{i=1}^n$ *be generated by a generalized linear model with sub-Gaussian Design, with* $\varepsilon_c$*-fraction of sample corruption and* $n = \Omega(\frac{d+\log(1/\delta)}{\varepsilon^2})$. *Assuming that* $C_0 \leq b''(\cdot) \leq C$ *for non-zero constants* $C_0, C$, $b(0) = 0, b'(0) = 0$, *and* $\log(c(y)) = O(\log(1/\varepsilon_c)), \forall y \leq \Theta(\sqrt{\log(1/\varepsilon_c)})$ *With probability* $1 - \delta$, *Algorithm 2 with parameters* $\varepsilon = \varepsilon_c, \eta = \varepsilon_c^2, R = \infty$ *terminate within* $\log(1/\varepsilon_c)/\varepsilon_c^2$ *iterations, and output an estimate* $\hat{\beta}$ *such that*

$$\|\hat{\beta} - \beta^*\| = O(\varepsilon_c \log(1/\varepsilon_c))$$

The proof is the same compared to the label corruption setting except that since $\mathbf{x}_i, i \in E$ is now controlled by the adversary, we can no longer bound $\sqrt{\frac{1}{n}\sum_{i \in \hat{S} \cap E}((\beta^* - \hat{\beta})^\top \mathbf{x}_i)^2}$ by the resilience property of (uncorrupted) sub-Gaussian samples. Instead, we will leverage the fact that corrupted sample with small covariance is also resilient.

## 6 Conclusion

In this paper, we provided a general theoretical analysis showing that a simple and practical heuristic namely the iterative trimmed MLE estimator achieves minimax optimal error rates upto a logarithmic factor under adversarial corruptions for a wide class of generalized linear models (GLMs). It would also be interesting to study whether our techniques can be extended to design robust algorithms for more general exponential families beyond GLMs.

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
