# A  Label Corruption Proofs

## A.1  Gaussian

*Proof of Theorem 4.2.* Applying Lemma A.12 with $\eta = \varepsilon^2$, $C = 1/\sigma^2$ implies the output of Algorithm 1 is a $O(\varepsilon/\sigma)$ approximate stationary point. Lemma A.1 then implies that $\|\hat{\beta} - \beta^*\| = O(\sigma\varepsilon\log(1/\varepsilon))$. To bound the number of iterations, we need an upper bound on the negative log-likelihood on $\beta = 0$, and a uniform lower bound on the negative log-likelihood. The initial negative log-likelihood is upper bounded by $\frac{1}{n}\sum_{i\in\hat{S}^{(1)}} y_i^2/\sigma^2$ where $S^{(1)}$ contains the smallest $(1 - \varepsilon)n$ labels, which is bounded by $O(\max(1, 1/\sigma^2))$. Trivially, there is a $0$ lower bound on the negative log-likelihood for Gaussian. Therefore, the algorithm will terminate in $\frac{1}{\min(1,\sigma^2)\varepsilon_c^2}$ iterations. $\qquad\square$

**Lemma A.1** (Approximate stationary point close to $\beta^*$ for Gaussian regression)**.** *Given a set of datapoints $S = \{\mathbf{x}_i, y_i\}_{i=1}^n$ generated by a Gaussian regression model with $\varepsilon$-fraction of label corruption. Let $\hat{\beta}$ be a $\varepsilon/\sigma$-stationary point defined in Definition D.4. Given that $n = \Omega(\frac{d+\log(1/\delta)}{\varepsilon^2})$, with probability $1 - \delta$, it holds that*

$$\|\hat{\beta} - \beta^*\| = O(\sigma\varepsilon\log(1/\varepsilon))$$

*Proof.* Let $\hat{S}$ be the set defined in Definition D.4. The first order stationary property guarantees

$$\frac{1}{n}\sum_{i\in\hat{S}}\frac{1}{\sigma^2}(y_i - \hat{\beta}^\top\mathbf{x}_i)(\beta^* - \hat{\beta})^\top\mathbf{x}_i \le \frac{\varepsilon}{\sigma}\|\beta^* - \hat{\beta}\|$$

Denote $T = G \setminus L$ as the uncorrupted set of data points. Then we get

$$\frac{1}{n}\sum_{i\in\hat{S}\cap T}(y_i - \hat{\beta}^\top\mathbf{x}_i)(\beta^* - \hat{\beta})^\top\mathbf{x}_i \le -\frac{1}{n}\sum_{i\in\hat{S}\cap E}(y_i - \hat{\beta}^\top\mathbf{x}_i)(\beta^* - \hat{\beta})^\top\mathbf{x}_i + \sigma\varepsilon\|\beta^* - \hat{\beta}\| \quad (1)$$

**Lower bound on the LHS**

We will first establish a lower bound on the LHS of Equation 1, which contains terms from $\hat{S}\cap T$. Note that

$$\frac{1}{n}\sum_{i\in\hat{S}\cap T}(y_i - \hat{\beta}^\top\mathbf{x}_i)(\beta^* - \hat{\beta})^\top\mathbf{x}_i$$

$$=\frac{1}{n}\sum_{i\in\hat{S}\cap T}(\eta_i + (\beta^* - \hat{\beta})^\top\mathbf{x}_i)(\beta^* - \hat{\beta})^\top\mathbf{x}_i$$

$$=\frac{1}{n}\sum_{i\in\hat{S}\cap T}\eta_i(\beta^* - \hat{\beta})^\top\mathbf{x}_i + ((\beta^* - \hat{\beta})^\top\mathbf{x}_i)^2$$

$$\overset{\text{resilience}}{\gtrsim} -\sigma\varepsilon\log(1/\varepsilon)\|\beta^* - \hat{\beta}\| + \|\beta^* - \hat{\beta}\|^2(1 - \varepsilon\log(1/\varepsilon)), \quad (2)$$

where we have leveraged the following resilience property in Equation 2

$$\|\frac{1}{n}\sum_{i\in\hat{S}\cap T}\eta_i\mathbf{x}_i\| \lesssim \sigma\varepsilon\log(1/\varepsilon)$$

$$\|\frac{1}{n}\sum_{i\in\hat{S}\cap T}\mathbf{x}_i\mathbf{x}_i^\top - I\| \lesssim \varepsilon\log(1/\varepsilon)$$

**Upper bound on the RHS**

Then we establish an upper bound on the RHS of Equation 1, which contains terms from $\hat{S} \cap E$

$$-\frac{1}{n} \sum_{i \in \hat{S} \cap E} (y_i - \hat{\beta}^\top \mathbf{x}_i)(\beta^* - \hat{\beta})^\top \mathbf{x}_i$$

$$\overset{\text{Cauchy-Schwarz}}{\lesssim} \left( \frac{1}{n} \sum_{i \in \hat{S} \cap E} (y_i - \hat{\beta}^\top \mathbf{x}_i)^2 \right)^{1/2} \left( \frac{1}{n} \sum_{i \in \hat{S} \cap E} ((\beta^* - \hat{\beta})^\top \mathbf{x}_i)^2 \right)^{1/2}$$

$$\overset{\text{resilience}}{\lesssim} \left( \frac{1}{n} \sum_{i \in \hat{S} \cap E} (y_i - \hat{\beta}^\top \mathbf{x}_i)^2 \right)^{1/2} \varepsilon^{1/2} \log(1/\varepsilon)^{1/2} \|\beta^* - \hat{\beta}\|$$

Observe that since

$$\left( \frac{1}{n} \sum_{i \in \hat{S}} (y_i - \hat{\beta}^\top \mathbf{x}_i)^2 \right) \overset{\text{optimality of } \hat{S}}{\leq} \left( \frac{1}{n} \sum_{i \in T} (y_i - \hat{\beta}^\top \mathbf{x}_i)^2 \right),$$

it holds that

$$\left( \frac{1}{n} \sum_{i \in \hat{S} \cap E} (y_i - \hat{\beta}^\top \mathbf{x}_i)^2 \right) \leq \left( \frac{1}{n} \sum_{i \in T} (y_i - \hat{\beta}^\top \mathbf{x}_i)^2 \right) - \left( \frac{1}{n} \sum_{i \in \hat{S} \cap T} (y_i - \hat{\beta}^\top \mathbf{x}_i)^2 \right)$$

$$= \left( \frac{1}{n} \sum_{i \in T \setminus \hat{S}} (y_i - \hat{\beta}^\top \mathbf{x}_i)^2 \right)$$

$$= \left( \frac{1}{n} \sum_{i \in T \setminus \hat{S}} (\eta_i + (\beta^* - \hat{\beta})^\top \mathbf{x}_i)^2 \right)$$

$$\overset{\text{resilience}}{=} O\left( \sigma^2 \varepsilon \log(1/\varepsilon) + \|\beta^* - \hat{\beta}\|^2 \varepsilon \log(1/\varepsilon) \right)$$

Combining we get

$$\frac{1}{n} \sum_{i \in \hat{S} \cap E} (y_i - \hat{\beta}^\top \mathbf{x}_i)(\beta^* - \hat{\beta})^\top \mathbf{x}_i \leq \varepsilon \log(1/\varepsilon)(\sigma \|\beta^* - \hat{\beta}\| + \|\beta^* - \hat{\beta}\|^2). \tag{3}$$

Plugging in Equation 2 and 3 into Equation 1, we we have

$$-\sigma \varepsilon \log(1/\varepsilon) \|\beta^* - \hat{\beta}\| + \|\beta^* - \hat{\beta}\|^2 (1 - \varepsilon \log(1/\varepsilon))$$

$$\lesssim \varepsilon \log(1/\varepsilon)(\sigma \|\beta^* - \hat{\beta}\| + \|\beta^* - \hat{\beta}\|^2) + \varepsilon \sigma \|\beta^* - \hat{\beta}\|$$

$$\implies \varepsilon \log(1/\varepsilon)(\sigma \|\beta^* - \hat{\beta}\| + \|\beta^* - \hat{\beta}\|^2) \gtrsim \|\beta^* - \hat{\beta}\|^2$$

$$\implies \|\beta^* - \hat{\beta}\| \lesssim \sigma \varepsilon \log(1/\varepsilon)$$

$\square$

## A.2  Poisson

**Lemma A.2** (Resilience condition for Poisson regression). *With probability* $0.99$ *it holds that for all* $Q \subset T$ *with* $|Q| \geq (1 - 2\varepsilon)n$,

$$\frac{1}{n} \| \sum_{i \in Q} \left( y_i - \exp(\beta^{*\top} \mathbf{x}_i) \right) \mathbf{x}_i \| \lesssim \varepsilon \exp(\Theta(\sqrt{\log(1/\varepsilon)}))$$

*and for all* $Q \subset T$ *with* $Q \leq \varepsilon n$,

$$\frac{1}{n} \sum_{i \in Q} y_i \lesssim \varepsilon \exp(\Theta(\sqrt{\log(1/\varepsilon)}))$$

$$\frac{1}{n} \| \sum_{i \in Q} y_i \mathbf{x}_i \| \lesssim \varepsilon \exp(\Theta(\sqrt{\log(1/\varepsilon)}))$$

*Proof.*

**Proof of the first statement**

We first prove the distribution of $\left(y - \exp(\beta^{*\top}\mathbf{x})\right)\mathbf{x}$ is $k$-th moment bounded. Note that the $k$-th moment along direction $v$ can be written as

$$\mathbb{E}[\left(y - \exp(\beta^{*\top}\mathbf{x})\right)^k (\mathbf{v}^\top\mathbf{x})^k]$$

$$\overset{\text{Cauchy-Schwarz}}{\leq} \sqrt{\mathbb{E}[\left(y - \exp(\beta^{*\top}\mathbf{x})\right)^{2k}]}\sqrt{\mathbb{E}[(\mathbf{v}^\top\mathbf{x})^{2k}]}$$

Applying the Poisson $k$-th moment bound from Fact D.5, and the $k$-th moment bound of sub-Gaussian random variable yields

$$\begin{aligned}
&\leq\sqrt{\mathbb{E}[\max\{\exp(2k(\beta^{*\top}\mathbf{x})),1\}2k^{2k}]}\sqrt{(2k)^k}\\
&\leq \exp(\Theta(k^2\|\beta^*\|^2))k^{\Theta(k)} \qquad\qquad\qquad\qquad\qquad (4)\\
&= \exp(\Theta(k^2)),
\end{aligned}$$

where we use the fact that $\mathbb{E}[\exp(\lambda x)] \leq \exp(\kappa^2\lambda^2)$ for $\kappa$-sub-Gaussian random variable in Equation 4. Applying Corollary G.1 in [ZJS19], we have that with probability 0.99, $\forall Q \subset T, |Q| \geq (1-2\varepsilon)n$,

$$\frac{1}{n}\|\sum_{i\in Q}(y_i - \exp(\langle\beta^*,\mathbf{x}_i\rangle))\mathbf{x}_i\| \leq \exp(Ck)(\varepsilon^{1-1/k} + \sqrt{d/n})$$

Setting $k = \sqrt{\log(1/\varepsilon)}/$ in the above upper bound yields an upper bound of

$$\varepsilon\exp(\Theta(\sqrt{\log(1/\varepsilon)}))$$

**Proof of the second statement**

Since $\mathbb{E}[y^k]^{1/k} = \exp(\Theta(k))$ Applying Corollary G.1 in [ZJS19] and setting $k = \sqrt{\log(1/\varepsilon)}$, we have that with probability 0.99, $\forall Q \subset T, |Q| \geq (1-\varepsilon)n$,

$$\frac{1}{n}|\sum_{i\in Q}(y_i - \mathbb{E}[y])| \leq \varepsilon\exp(\Theta(\sqrt{\log(1/\varepsilon)}))$$

This implies for all $Q \subset T$ with $|Q| \leq \varepsilon n$,

$$\frac{1}{n}|\sum_{i\in Q}y_i - \mathbb{E}[y]| \leq \varepsilon\exp(\Theta(\sqrt{\log(1/\varepsilon)}))$$

$$\overset{\mathbb{E}[y]=O(1)}{\Longrightarrow}\frac{1}{n}|\sum_{i\in Q}y_i| \leq \varepsilon\exp(\Theta(\sqrt{\log(1/\varepsilon)}))$$

Setting $k = \sqrt{\log(1/\varepsilon)}/$ in the above upper bound yields an upper bound of $\varepsilon\exp(\Theta(\sqrt{\log(1/\varepsilon)}))$

**Proof of the third statement**

Since we have the same $k$-th moment bound on $y$ and $y - \mathbb{E}[y]$. The bound $\frac{1}{n}\|\sum_{i\in Q}y_i\mathbf{x}_i\| \leq \varepsilon\exp(\Theta(\sqrt{\log(1/\varepsilon)}))$ can be proved similarly. $\qquad\square$

**Theorem A.3** (Poisson regression with label corruption (Restatement of Theorem 4.3)). *Let $S = \{\mathbf{x}_i, y_i\}_{i=1}^n$ be a set of data points generated by a Poisson regression model with sub-Gaussian design, with $\varepsilon_c$-fraction of label corruption and $n = \Omega(\frac{d}{\varepsilon_c^2})$. With probability 0.99, Algorithm 1 with parameters $\varepsilon = 2\varepsilon_c, \eta = \varepsilon_c^2/(dn), R \geq \|\beta^*\|$ terminate within $dn/\varepsilon_c^2$ iterations, and output an estimate $\hat{\beta}$ such that*

$$\|\hat{\beta} - \beta^*\| \lesssim \varepsilon_c\exp(\Theta(\sqrt{\log(1/\varepsilon_c)}))$$

*Proof of Theorem A.3.* Lemma A.13 implies the output of Algorithm 1 is a $\max(\varepsilon, \frac{\varepsilon^2}{\|\beta^* - \hat{\beta}\|})$ approximate stationary point. Lemma A.4 then implies that $\|\hat{\beta} - \beta^*\| = O(\varepsilon \exp(\sqrt{\log(1/\varepsilon)}))$. To bound the number of iterations, we need an upper bound on the negative log-likelihood on $\beta = 0$, and a uniform lower bound on the negative log-likelihood. The initial negative log-likelihood is upper bounded by $\frac{1}{n} \sum_{i \in \hat{S}^{(1)}} \log(y_i!) + 1 \leq O(\mathbb{E}[y_i^2] + 1) = O(1)$ where $S^{(1)}$ contains the smallest $(1 - \varepsilon)n$ labels. Trivially, there is a 0 lower bound on the negative log-likelihood for Poisson distribution. Therefore, the algorithm will terminate in $dn/\varepsilon_c^2$ iterations. □

**Lemma A.4** (Approximate stationary point close to $\beta^*$ for Poisson regression (Restatement of Lemma 4.7))**.** *Given a set of datapoints $S = \{\mathbf{x}_i, y_i\}_{i=1}^n$ generated by a Poisson regression model with $\varepsilon$-fraction of label corruption, and the largest $\varepsilon n$ labels removed. Let $\hat{\beta}$ be a $\max(\varepsilon, \varepsilon^2/\|\beta^* - \hat{\beta}\|)$-stationary point defined in Definition D.4 and $\|\hat{\beta}\| \leq R$. Given that $n = \Omega(\frac{d}{\varepsilon^2})$, with probability 0.99, it holds that*

$$\|\hat{\beta} - \beta^*\| = O(\varepsilon \exp(\Theta(\sqrt{\log(1/\varepsilon)})))$$

*Proof.* Recall that $y_i \sim \text{Poi}(\exp({\beta^*}^\top \mathbf{x}_i))$. Poisson regression log-likelihood:

$$\log \Pr(y_i | \langle \beta^*, \mathbf{x}_i \rangle) = y_i ({\beta^*}^\top \mathbf{x}_i) - \exp({\beta^*}^\top \mathbf{x}_i) - \log y_i!$$

If $\|\beta^* - \hat{\beta}\| \leq \varepsilon$, the proof is done. When $\|\beta^* - \hat{\beta}\| \geq \varepsilon$, the first order approximate stationary property guarantees

$$\frac{1}{n} \sum_{i \in S} (y_i - \exp(\hat{\beta}^\top \mathbf{x}_i))(\beta^* - \hat{\beta})^\top \mathbf{x}_i \leq \varepsilon \|\beta^* - \hat{\beta}\|$$

$$\implies \frac{1}{n} \sum_{i \in \hat{S} \cap T} (y_i - \exp(\hat{\beta}^\top \mathbf{x}_i))(\beta^* - \hat{\beta})^\top \mathbf{x}_i$$

$$\leq -\frac{1}{n} \sum_{i \in \hat{S} \cap E} (y_i - \exp(\hat{\beta}^\top \mathbf{x}_i))(\beta^* - \hat{\beta})^\top \mathbf{x}_i + \varepsilon \|\beta^* - \hat{\beta}\|$$

**Lower bound on the LHS**

We will first establish a lower bound on the LHS, which contains terms from $\hat{S} \cap T$. Note that

$$\frac{1}{n} \sum_{i \in \hat{S} \cap T} (y_i - \exp(\hat{\beta}^\top \mathbf{x}_i))(\beta^* - \hat{\beta})^\top \mathbf{x}_i$$

$$= \frac{1}{n} \sum_{i \in \hat{S} \cap T} \left( y_i - \exp({\beta^*}^\top \mathbf{x}_i) \right) (\beta^* - \hat{\beta})^\top \mathbf{x}_i \tag{5}$$

$$+ \frac{1}{n} \sum_{i \in \hat{S} \cap T} \left( \exp({\beta^*}^\top \mathbf{x}_i) - \exp(\hat{\beta}^\top \mathbf{x}_i) \right) (\beta^* - \hat{\beta})^\top \mathbf{x}_i, \tag{6}$$

and we bound the two terms separately.

Lemma A.2 implies

$$\frac{1}{n} \sum_{i \in \hat{S} \cap T} \left( y_i - \exp(\beta^\top \mathbf{x}_i) \right) (\beta^* - \hat{\beta})^\top \mathbf{x}_i \leq \|\hat{\beta} - \beta^*\| \varepsilon \exp(\Theta(\sqrt{\log(1/\varepsilon)})) \tag{7}$$

Now we bound the second term in Equation 6. By resilience property (Proposition D.1), the set $L_\beta = \{\beta^\top \mathbf{x}_i < -C\|\beta\| \log(1/\gamma)\}$ satisfies $|L_\beta| \leq \gamma n$ for any $\gamma > \varepsilon$, and also it is clear that $\frac{1}{n} \sum_{i \in (\hat{S} \cap T) \setminus L_\beta} \mathbf{x}_i \mathbf{x}_i^\top \succeq (1 - C\gamma \log(1/\gamma)) \cdot I$, hence

$$\frac{1}{n} \sum_{i \in (\hat{S} \cap T) \setminus L_\beta} \exp(\beta^\top \mathbf{x}_i) \mathbf{x}_i \mathbf{x}_i^\top \succeq \exp(-C\|\beta\| \log(1/\gamma))(1 - C\gamma \log(1/\gamma)) \cdot I$$

Let $\gamma$ be a small constant yields

$$\frac{1}{n} \sum_{i \in \hat{S} \cap T} \exp(\beta^\top \mathbf{x}_i) \mathbf{x}_i \mathbf{x}_i^\top \succeq e^{-O(\max(\|\beta\|, 1))} I$$

This implies that $\sum_{i \in \hat{S} \cap T} \exp(\beta^\top \mathbf{x}_i)$ is a strongly convex function in $\beta$ in the range of $\|\beta\| = O(1)$. Since both $\|\beta^*\| = O(1)$ and $\|\hat{\beta}\| = O(1)$, by the definition of strongly convex function

$$\frac{1}{n}(\nabla \sum_{i \in \hat{S} \cap T} \exp(\beta^{*\top} \mathbf{x}_i) - \nabla \sum_{i \in \hat{S} \cap T} \exp(\hat{\beta}^\top \mathbf{x}_i))(\beta^* - \hat{\beta}) \geq \Omega(\|\beta^* - \hat{\beta}\|^2)$$

$$\implies \frac{1}{n}(\sum_{i \in \hat{S} \cap T} \exp(\beta^{*\top} \mathbf{x}_i) - \sum_{i \in \hat{S} \cap T} \exp(\hat{\beta}^\top \mathbf{x}_i))(\beta^* - \hat{\beta})^\top \mathbf{x}_i \geq \Omega(\|\beta^* - \hat{\beta}\|^2)$$

Combining this term with Equation 7, we have shown that

$$\frac{1}{n} \sum_{i \in \hat{S} \cap T} (y_i - \exp(\hat{\beta}^\top \mathbf{x}_i))(\beta - \hat{\beta})^\top \mathbf{x}_i \geq C_1 \|\beta^* - \hat{\beta}\|^2 - \|\hat{\beta} - \beta^*\| \varepsilon \exp(C_2 \sqrt{\log(1/\varepsilon)}) \quad (8)$$

**Upper bound on the RHS**

**1. Upper bound on the negative log-likelihood.** Recall that $|\hat{S}| = (1 - 2\varepsilon)n$, $|T| = (1 - \varepsilon)n$, and the negative log-likelihood of Poisson regression is

$$\exp(\hat{\beta}^\top \mathbf{x}_i) - y_i(\hat{\beta}^\top \mathbf{x}_i) + \log y_i!.$$

By the optimality of $\hat{S}$, it must hold that

$$\max_{i \in \hat{S} \cap E} \exp(\hat{\beta}^\top \mathbf{x}_i) - y_i(\hat{\beta}^\top \mathbf{x}_i) + \log y_i!$$

$$\leq \min_{i \in T \setminus \hat{S}} \exp(\hat{\beta}^\top \mathbf{x}_i) - y_i(\hat{\beta}^\top \mathbf{x}_i) + \log y_i$$

since otherwise one can replace a data point in $\hat{S} \cap E$ by one in $T \setminus \hat{S}$. Since $|T \setminus \hat{S}| \geq \varepsilon n$

$$\min_{i \in T \setminus \hat{S}} \exp(\hat{\beta}^\top \mathbf{x}_i) - y_i(\hat{\beta}^\top \mathbf{x}_i) + \log y_i$$

$$\leq \max_{Q \subset T, |Q| = \varepsilon n} \min_{i \in Q} \exp(\hat{\beta}^\top \mathbf{x}_i) - y_i(\hat{\beta}^\top \mathbf{x}_i) + \log y_i!$$

$$\leq \left( \max_{Q \subset T, |Q| = \varepsilon n/3} \min_{i \in Q} \exp(\hat{\beta}^\top \mathbf{x}_i) + \max_{Q \subset T, |Q| = \varepsilon n/3} \min_{i \in Q} -y_i(\hat{\beta}^\top \mathbf{x}_i) + \max_{Q \subset T, |Q| = \varepsilon n/3} \min_{i \in Q} \log y_i! \right)$$

Applying Lemma A.2 and Proposition D.1 gives

$$\leq \left( \exp(\Theta(\sqrt{\log(1/\varepsilon)})) + \max_{Q \subset T, |Q| = \varepsilon n/3} \min_{i \in Q} y_i \log y_i \right)$$

$$\leq \left( \exp(\Theta(\sqrt{\log(1/\varepsilon)})) + \exp(\Theta(\sqrt{\log(1/\varepsilon)})) \Theta(\sqrt{\log(1/\varepsilon)}) \right)$$

$$\leq \left( \exp(\Theta(\sqrt{\log(1/\varepsilon)})) \right)$$

**2. Turn likelihood bound into square error bound**. Now we have an upper bound on the negative log-likelihood of a data point in $\hat{S} \cap E$, next step we will turn the log-likelihood bound to a squared error bound. Define proxy function $g_{\hat{\beta}^\top \mathbf{x}_i}(y_i)$ as

$$g_{\hat{\beta}^\top \mathbf{x}_i}(y_i) = \exp(\hat{\beta}^\top \mathbf{x}_i) - y_i(\hat{\beta}^\top \mathbf{x}_i) + y_i \log y_i - y_i,$$

which, by Fact D.6, is always smaller the negative log-likelihood function

$$-\log f(y_i | \langle \hat{\beta}, \mathbf{x}_i \rangle) = \exp(\hat{\beta}^\top \mathbf{x}_i) - y_i(\hat{\beta}^\top \mathbf{x}_i) + \log y_i!,$$

and hence $g_{\hat{\beta}^\top \mathbf{x}_i}(y_i) \leq \exp(\Theta(\sqrt{\log(1/\varepsilon)}))$.

Note that

$$g_{\hat{\beta}^\top \mathbf{x}_i}(\exp(\hat{\beta}^\top \mathbf{x}_i)) = 0$$

$$g'_{\hat{\beta}^\top \mathbf{x}_i}(\exp(\hat{\beta}^\top \mathbf{x}_i)) = 0$$

$$g'_{\hat{\beta}^\top \mathbf{x}_i}(y_i) = \frac{1}{y_i}.$$

Therefore, we can lower bound $g'_{\hat{\beta}^\top \mathbf{x}_i}(y_i)$ as

$$g_{\hat{\beta}^\top \mathbf{x}_i}(y_i) \geq \min(\frac{1}{y_i}, \frac{1}{\exp(\hat{\beta}^\top \mathbf{x}_i)})(y_i - \exp(\hat{\beta}^\top \mathbf{x}_i))^2.$$

Now there are two cases to consider: 1) if $y_i \geq \exp(\hat{\beta}^\top \mathbf{x}_i)/2$, it holds that

$$(y_i - \exp(\hat{\beta}^\top \mathbf{x}_i))^2 \leq \sqrt{y_i g_{\hat{\beta}^\top \mathbf{x}_i}(y_i)} \leq 2\exp(\Theta(\sqrt{\log(1/\varepsilon)})),$$

where we leveraged the fact that $g_{\hat{\beta}^\top \mathbf{x}_i}(y_i) \leq \exp(\Theta(\sqrt{\log(1/\varepsilon)}))$ and $y_i \leq \exp(\Theta(\sqrt{\log(1/\varepsilon)}))$ after throwing away the largest $\varepsilon n$ $y_i$s in the beginning of the algorithm. 2) if $y < \exp(\hat{\beta}^\top \mathbf{x}_i)/2$, we have

$$g_{\hat{\beta}^\top \mathbf{x}_i}(y_i) \geq \frac{1}{\exp(\hat{\beta}^\top \mathbf{x}_i)}(y_i - \exp(\hat{\beta}^\top \mathbf{x}_i))^2 \geq \frac{1}{2}|y_i - \exp(\hat{\beta}^\top \mathbf{x}_i)|$$

$$\implies (y_i - \exp(\hat{\beta}^\top \mathbf{x}_i))^2 \leq \exp(\Theta(\sqrt{\log(1/\varepsilon)})).$$

By Cauchy Schwarz

$$-\frac{1}{n}\sum_{i\in\hat{S}\cap E}(y_i - \exp(\hat{\beta}^\top \mathbf{x}_i))(\beta^* - \hat{\beta})^\top \mathbf{x}_i$$

$$\leq \sqrt{\frac{1}{n}\sum_{i\in\hat{S}\cap E}(y_i - \exp(\hat{\beta}^\top \mathbf{x}_i))^2}\sqrt{\frac{1}{n}\sum_{i\in\hat{S}\cap E}((\beta^* - \hat{\beta})^\top \mathbf{x}_i)^2}$$

$$\leq \varepsilon^{1/2}\exp\left(\Theta(\sqrt{\log(1/\varepsilon)})\right)\varepsilon^{1/2}\log^{1/2}(1/\varepsilon)\|\beta^* - \hat{\beta}\|$$

$$= \varepsilon\exp\left(C_3(\sqrt{\log(1/\varepsilon)})\right)\|\beta^* - \hat{\beta}\| \qquad (9)$$

**Combining LHS and RHS**. Combining Equation 8 and Equation 9 yields

$$C_1\|\beta^* - \hat{\beta}\|^2 - \|\hat{\beta} - \beta^*\|\varepsilon\exp(C_2(\sqrt{\log(1/\varepsilon)}))$$

$$\leq \varepsilon\exp\left(C_3(\sqrt{\log(1/\varepsilon)})\right)\|\beta^* - \hat{\beta}\| + \varepsilon\|\beta^* - \hat{\beta}\|$$

$$\implies \|\beta^* - \hat{\beta}\| \leq \varepsilon\exp\left(C_4(\sqrt{\log(1/\varepsilon)})\right)$$

$\square$

## A.3 Binomial

*Proof of Theorem 4.4.* Lemma A.12 implies the output of Algorithm 1 is a $\max(\varepsilon/\sqrt{m}, \frac{\varepsilon^2}{m\|\beta^* - \hat{\beta}\|})$ approximate stationary point. Lemma A.5 then implies that $\|\hat{\beta} - \beta^*\| = O(\varepsilon\sqrt{\frac{\log(m/\varepsilon)\log(1/\varepsilon)}{m}})$. The initial negative log-likelihood is upper bounded by $m$, and trivially, there is a 0 lower bound on the negative log-likelihood. Therefore, the algorithm will terminate in $m^2/\varepsilon_c^2$ iterations. $\square$

**Lemma A.5** (Approximate stationary point close to $\beta^*$ for Binomial regression). *Given a set of datapoints $S = \{\mathbf{x}_i, y_i\}_{i=1}^n$ generated by a Binomial regression model with $m$ trials and $\varepsilon$-fraction of label corruption. Let $\hat{\beta}$ be an $\max(\varepsilon/\sqrt{m}, \frac{\varepsilon^2}{m\|\beta^* - \hat{\beta}\|})$-stationary point defined in Definition D.4 with $\|\hat{\beta}\| \leq R$. Given that $n = \Omega(\frac{d + \log(1/\delta)}{\varepsilon^2})$, with probability $1 - \delta$, it holds that*

$$\|\beta^* - \hat{\beta}\| \leq O(\varepsilon\sqrt{\frac{\log(m/\varepsilon)\log(1/\varepsilon)}{m}})$$

*Proof.* Recall that the log-likelihood of Binomial regression is

$$\log \Pr(y|\langle \beta, \mathbf{x} \rangle) = y\beta^\top \mathbf{x} - m \log(1 + \exp(\beta^\top \mathbf{x})) + \log \binom{m}{y}$$

The following holds for fucntion $b(\beta^\top \mathbf{x})$ in the Binomial regression.

$$b(\beta^\top \mathbf{x}) = m \log(1 + \exp(\beta^\top \mathbf{x}))$$

$$\text{Mean function: } b'(\beta^\top \mathbf{x}) = m \frac{1}{1 + \exp(-\beta^\top \mathbf{x})}$$

$$\text{Variance function: } b''(\beta^\top \mathbf{x}) = m \frac{1}{(1 + \exp(-\beta^\top \mathbf{x}))(1 + \exp(\beta^\top \mathbf{x}))}$$

The first order approximate stationary property guarantees

$$\frac{1}{n} \sum_i (y_i - b'(\hat{\beta}^\top \mathbf{x}_i))(\beta^* - \hat{\beta})^\top \mathbf{x}_i \leq \frac{\varepsilon}{\sqrt{m}} \|\beta^* - \hat{\beta}\|$$

$$\implies \frac{1}{n} \sum_{i \in \hat{S} \cap T} (y_i - b'(\hat{\beta}^\top \mathbf{x}_i))(\beta^* - \hat{\beta})^\top \mathbf{x}_i$$

$$\leq -\frac{1}{n} \sum_{i \in \hat{S} \cap E} (y_i - b'(\hat{\beta}^\top \mathbf{x}_i))(\beta^* - \hat{\beta})^\top \mathbf{x}_i + \frac{\varepsilon}{\sqrt{m}} \|\beta^* - \hat{\beta}\|$$

**Lower bound on the LHS**

We will first establish a lower bound on the LHS, which contains terms from $\hat{S} \cap T$. Note that

$$\sum_{i \in \hat{S} \cap T} (y_i - b'(\hat{\beta}^\top \mathbf{x}_i))(\beta^* - \hat{\beta})\mathbf{x}_i$$

$$= \sum_{i \in \hat{S} \cap T} \left( y_i - b'(\beta^{*\top} \mathbf{x}_i) \right)(\beta^* - \hat{\beta})^\top \mathbf{x}_i + \sum_{i \in \hat{S} \cap T} \left( b'(\beta^{*\top} \mathbf{x}_i) - b'(\hat{\beta}^\top \mathbf{x}_i) \right)(\beta^* - \hat{\beta})^\top \mathbf{x}_i, \quad (10)$$

and we bound the two terms separately. Note that $\frac{y_i - b'(\beta^\top \mathbf{x}_i)}{\sqrt{m}}$ has sub-Gaussian norm 1, $\mathbf{x}_i$ has sub-Gaussian norm 1. Let

$$\tilde{x}_i = \begin{bmatrix} \mathbf{x}_i \\ (y_i - b'(\beta^\top \mathbf{x}_i))/\sqrt{m} \end{bmatrix}$$

which is a 1 sub-Gaussian random vector with mean 0 and covariance $\tilde{\Sigma} = \begin{bmatrix} I_d & 0 \\ 0 & c \end{bmatrix}$ for a constant $c$.
By Proposition D.1 we have that

$$\left\| \frac{1}{n} \sum_{i \in \hat{S} \cap T} \tilde{\mathbf{x}}_i \tilde{\mathbf{x}}_i^\top - \tilde{\Sigma} \right\| \leq \varepsilon \log(1/\varepsilon).$$

which implies for $\mathbf{u} = [0, \ldots, 1]^\top \in \mathbb{R}^{d+1}$ and any $\mathbf{w} = [\mathbf{v}, 0]^\top, \|\mathbf{v}\| = 1, \mathbf{v} \in \mathbb{R}^d$

$$\frac{1}{n\sqrt{m}} \sum_{i \in \hat{S} \cap T} \left( y_i - b'(\beta^{*\top} \mathbf{x}_i) \right) \mathbf{v}^\top \mathbf{x}_i$$

$$= \frac{1}{n} \sum_{i \in \hat{S} \cap T} \mathbf{u}^\top \tilde{\mathbf{x}}_i \tilde{\mathbf{x}}_i^\top \mathbf{w} \leq \varepsilon \log(1/\varepsilon).$$

Hence,

$$\frac{1}{n} \sum_{i \in \hat{S} \cap T} \left( y_i - b'(\beta^{*\top} \mathbf{x}_i) \right)(\beta^* - \hat{\beta})^\top \mathbf{x}_i \leq \sqrt{m}\varepsilon \log(1/\varepsilon)\|\beta^* - \hat{\beta}\| \tag{11}$$

Now we bound the second term in Equation 10. By resilience property of sub-Gaussian distribution, for any $\|\beta\| = O(1)$, there is a constant $C$ such that the set $L_\beta = \{|\beta^\top \mathbf{x}_i| > C \log(1/\gamma)\}$ satisfies $|L_\beta| \leq \gamma n$ for any $\gamma > \varepsilon$, and also $\sum_{i \in (\hat{S} \cap T) \setminus L_\beta} \mathbf{x}_i \mathbf{x}_i^\top \succeq (1 - C\gamma \log(1/\gamma)) \cdot I$. Hence

$$\sum_{i\in(\hat{S}\cap T)\setminus L_\beta} b''(\beta^\top \mathbf{x}_i)\mathbf{x}_i\mathbf{x}_i^\top \succeq m\frac{1}{(1+1/\gamma^C)(1+\gamma^C)}(1 - C\gamma\log(1/\gamma))I \succeq \Theta(m\cdot I)$$

by Setting $\gamma$ as a small constant. This implies $\sum_{i\in(\hat{S}\cap T)\setminus L_\beta} b(\beta^\top \mathbf{x}_i)$ is a strongly convex function with parameter $\Theta(m)$ when $\|\beta\| = O(1)$. By the definition of strongly convex function, we have

$$\sum_{i\in\hat{S}\cap T}\left(b'(\beta^\top \mathbf{x}_i) - b'(\hat{\beta}^\top \mathbf{x}_i)\right)(\beta - \hat{\beta})^\top \mathbf{x}_i \geq \Omega(m\|\beta - \hat{\beta}\|^2)$$

Combining the two terms, we have shown that

$$\frac{1}{n}\sum_{i\in\hat{S}\cap T}(y_i - b'(\hat{\beta}^\top \mathbf{x}_i))(\beta^* - \hat{\beta})^\top \mathbf{x}_i \geq C_1(m\|\beta^* - \hat{\beta}\|^2) - C_2(\sqrt{m}\|\hat{\beta} - \beta^*\|\varepsilon\log(1/\varepsilon)) \quad (12)$$

**Upper bound on the RHS**

**1. Upper bound on the negative log-likelihood.** By the optimality of $\hat{S}$, it must hold that

$$\sum_{i\in\hat{S}\cap E} b(\hat{\beta}^\top \mathbf{x}_i) - y_i(\hat{\beta}^\top \mathbf{x}_i) - \log\binom{m}{y_i} \leq \sum_{i\in T\setminus\hat{S}} b(\hat{\beta}^\top \mathbf{x}_i) - y_i(\hat{\beta}^\top \mathbf{x}_i) - \log\binom{m}{y_i}$$

$$\leq \sum_{i\in T\setminus\hat{S}} b(\beta^{*\top} \mathbf{x}_i) - y_i(\beta^{*\top} \mathbf{x}_i) - \log\binom{m}{y_i}$$

$$+ \sum_{i\in T\setminus\hat{S}} (b(\hat{\beta}^\top \mathbf{x}_i) - y_i(\hat{\beta}^\top \mathbf{x}_i)) - (b(\beta^{*\top} \mathbf{x}_i) - y_i(\beta^{*\top} \mathbf{x}_i))$$

The first summation corresponds to the negative log-likelihood of good data under the right model $\beta^*$. The second summation corresponds to the shift in the likelihood from $\beta^*$ to $\hat{\beta}$. We first bound the first summation.

Define random variable $z_i = -\log f(y_i|\langle\beta^*, \mathbf{x}_i\rangle) = b(\hat{\beta}^\top \mathbf{x}_i) - y_i(\hat{\beta}^\top \mathbf{x}_i) - \log\binom{m}{y_i}$. Notice that condition on $\mathbf{x}_i$, $z_i$ can only take $m$ values, hence it is easy to see that $\forall\delta, \Pr(z_i \geq \log(1/\delta)|\mathbf{x}_i) \leq m\delta$. Since this is true regardless of $\mathbf{x}_i$, we have

$$\Pr(z_i \geq \log(1/\delta)) \leq m\delta$$
$$\implies \Pr(z_i - \log m \geq t) \leq e^{-t}$$

Hence $z_i - \log m$ is a 1 sub-exponential random variable. By the resilience property (Corollary D.2), we have that

$$\frac{1}{n}\sum_{i\in T\setminus\hat{S}} b(\beta^{*\top} \mathbf{x}_i) - y_i(\beta^{*\top} \mathbf{x}_i) - \log\binom{m}{y_i} = \frac{1}{n}\sum_{i\in T\setminus\hat{S}} z_i \lesssim \varepsilon\log(m/\varepsilon).$$

Now we bound the second summation. Note that

$$b(\hat{\beta}^\top \mathbf{x}_i) - y_i(\hat{\beta}^\top \mathbf{x}_i)$$
$$\leq b(\beta^{*\top} \mathbf{x}_i) - y_i(\beta^{*\top} \mathbf{x}_i) + (b'(\beta^{*\top} \mathbf{x}_i) - y_i)(\hat{\beta} - \beta^*)^\top \mathbf{x}_i + \max b''(\beta^\top \mathbf{x}_i)\cdot((\beta^* - \hat{\beta})^\top \mathbf{x}_i)^2$$

Leveraging Equation 11, resilience of sub-Gaussian sample (Proposition D.1), and $b''(\beta^\top \mathbf{x}_i) \leq m$, we get

$$\frac{1}{n}\sum_{i\in T\setminus\hat{S}} b(\hat{\beta}^\top) - y_i(\hat{\beta}^\top \mathbf{x}_i) - \sum_{i\in T\setminus\hat{S}} b(\beta^{*\top} \mathbf{x}_i) - y_i(\beta^{*\top} \mathbf{x}_i))$$

$$\lesssim \sqrt{m}\varepsilon\log(1/\varepsilon)\|\beta^* - \hat{\beta}\| + m\varepsilon\log(1/\varepsilon)\|\beta^* - \hat{\beta}\|^2$$

Combining the two summations yields

$$\frac{1}{n} \sum_{i \in \hat{S} \cap E} b(\hat{\beta}^\top \mathbf{x}_i) - y_i(\hat{\beta}^\top \mathbf{x}_i) - \log \binom{m}{y_i}$$

$$\leq \frac{1}{n} \sum_{i \in T \setminus \hat{S}} b(\hat{\beta}^\top \mathbf{x}_i) - y_i(\hat{\beta}^\top \mathbf{x}_i) - \log \binom{m}{y_i}$$

$$\lesssim \sqrt{m}\varepsilon \log(1/\varepsilon)\|\beta^* - \hat{\beta}\| + m\varepsilon \log(1/\varepsilon)\|\beta^* - \hat{\beta}\|^2 + \varepsilon \log(m/\varepsilon)$$

**2. Turn likelihood bound into square error bound**. From Fact D.7, define proxy function

$$g_{\hat{\beta}^\top \mathbf{x}_i}(y_i) = b(\hat{\beta}^\top \mathbf{x}_i) - y_i(\hat{\beta}^\top \mathbf{x}_i) + y_i \log(\frac{y_i}{m}) + (m - y_i) \log(\frac{m - y_i}{m}) + \log \frac{y(m - y)}{m} + C$$

$$\leq b(\hat{\beta}^\top \mathbf{x}_i) - y_i(\hat{\beta}^\top \mathbf{x}_i) - \log \binom{m}{y_i}$$

Note that

$$g_{\hat{\beta}^\top \mathbf{x}_i}(b'(\hat{\beta}^\top \mathbf{x}_i)) = C$$

$$g'_{\hat{\beta}^\top \mathbf{x}_i}(b'(\hat{\beta}^\top \mathbf{x}_i)) = 0$$

$$g'_{\hat{\beta}^\top \mathbf{x}_i}(y_i) = \frac{1}{y_i} + \frac{1}{m - y_i} \geq 4/m.$$

Combining this with the likelihood bound we get

$$\frac{1}{n} \sum_{i \in \hat{S} \cap E} (y_i - b'(\hat{\beta}^\top \mathbf{x}_i))^2 \lesssim \varepsilon \left( m^2 \left( \|\beta^* - \hat{\beta}\|^2 + \sqrt{1/m}\|\beta^* - \hat{\beta}\| \right) \log(1/\varepsilon) + (\log m/\varepsilon)m \right)$$

By Cauchy Schwaz

$$-\frac{1}{n} \sum_{i \in \hat{S} \cap E} (y_i - b'(\hat{\beta}^\top \mathbf{x}_i))(\beta - \hat{\beta})^\top \mathbf{x}_i \lesssim \sqrt{\frac{1}{n} \sum_{i \in \hat{S} \cap E} (y_i - b(\hat{\beta}^\top \mathbf{x}_i))^2} \sqrt{\frac{1}{n} \sum_{i \in \hat{S} \cap E} ((\beta - \hat{\beta})^\top \mathbf{x}_i)^2}$$

$$\lesssim \varepsilon m \left( \sqrt{\|\beta^* - \hat{\beta}\|^2 + \frac{1}{\sqrt{m}}\|\beta^* - \hat{\beta}\| \log(1/\varepsilon)} + \sqrt{\frac{\log(m/\varepsilon)\log(1/\varepsilon)}{m}} \right) \cdot \|\beta^* - \hat{\beta}\|$$

$$\tag{13}$$

Combining Equation 12 and Equation 13 yields

$$\|\beta^* - \hat{\beta}\| \leq O(\varepsilon \sqrt{\frac{\log(m/\varepsilon)\log(1/\varepsilon)}{m}})$$

$\square$

## A.4 Generalized Linear Model

**Theorem A.6** (Generalized linear model with label corruption (Restatement of Theorem 4.5)). *Let $S = \{\mathbf{x}_i, y_i\}_{i=1}^n$ be generated by a generalized linear model with sub-Gaussian Design, with $\varepsilon_c$-fraction of label corruption and $n = \Omega(\frac{d + \log(1/\delta)}{\varepsilon^2})$. Assuming that $C_0 \leq b''(\cdot) \leq C$ for non-zero constants $C_0, C$, $b(0) = 0, b'(0) = 0$, and $\log(c(y)) = O(\log(1/\varepsilon)), \forall y \leq \Theta(\sqrt{\log(1/\varepsilon)})$ With probability $1 - \delta$, Algorithm 1 with parameters $\varepsilon = \varepsilon_c, \eta = \varepsilon_c^2, R = \infty$ terminate within $\log(1/\varepsilon_c)/\varepsilon_c^2$ iterations, and output an estimate $\hat{\beta}$ such that*

$$\|\hat{\beta} - \beta^*\| = O(\varepsilon_c \log(1/\varepsilon_c))$$

*Proof.* Lemma A.12 implies the output of Algorithm 1 is a $\max(\varepsilon_c, \frac{\varepsilon_c^2}{\|\beta^* - \hat{\beta}\|})$ approximate stationary point. Lemma A.8 then implies that $\|\hat{\beta} - \beta^*\| = O(\varepsilon_c \log(1/\varepsilon_c))$. Now we analyze the

iteration complexity. Note that Equation 17 shows that the $\varepsilon$ quantile of negative log-likelihood $-\log f(y_i|\langle \beta^*, \mathbf{x}_i \rangle)$ is upper bounded by $\log(1/\varepsilon)$. Since the algorithm start $\hat{\beta}$ from 0, using the fact that $b(0) = 0$, the initial negative log-likelihood is upper bounded by

$$-\log f(y_i|\langle \beta^*, \mathbf{x}_i \rangle) - b(\langle \beta^*, \mathbf{x}_i \rangle) + y_i \langle \beta^*, \mathbf{x}_i \rangle$$
$$\leq -\log f(y_i|\langle \beta^*, \mathbf{x}_i \rangle) + y_i \langle \beta^*, \mathbf{x}_i \rangle.$$

Since $y_i \leq \sqrt{\log(1/\varepsilon_c)}$ and the $\varepsilon_c$-quantile of $\langle \beta^*, \mathbf{x} \rangle$ is bounded by $\sqrt{\log(1/\varepsilon_c)}$. We get that the negative log-likelihood is upper bounded by $\Theta(\log(1/\varepsilon_C))$. From Equation 18, we know that there is a $-\log(1/\varepsilon_c)$ lower bound on the negative log-likelihood. Therefore, the algorithm will terminate in $\log(1/\varepsilon_c)/\varepsilon_c^2$ iterations. $\qquad\square$

**Remark A.7.** *The assumption that $b''(\cdot) \leq C$ makes sure the variance of $y$ is bounded, and $b''(\cdot) \geq C_0$ makes sure the distribution of $y$ never degenerate to a singular point. Without these conditions, in the minimax sense, learning $\beta^*$ with finite sample is impossible. The additional condition on $c(y)$ exists to rule out having a single $y_i$ with extremely large density. This would be problematic since the adversary can inject $y_i$ at the point and the trimmed maximum likelihood estimator will not be able to remove these datapoints. Note that this is a mild condition since this essentially only requires $c(y) \leq \exp(y^2)$, and it is trivially true for probability mass function. All these assumptions are satisfied by the Gaussian regression model studied in this work.*

**Lemma A.8** (Approximate stationary point close to $\beta^*$ for generalized linear regression)**.** *Given a set of datapoints $S = \{\mathbf{x}_i, y_i\}_{i=1}^n$ generated by a generalized linear model with $\varepsilon$-fraction of label corruption. Let $\hat{\beta}$ be an $\max(\varepsilon, \varepsilon^2/\|\beta^* - \hat{\beta}\|)$-stationary point defined in Definition D.4. Assuming that $C_0 \leq b''(\cdot) \leq C$ for non-zero constants $C_0, C$, $b(0) = 0, b'(0) = 0$, and $\log(c(y)) = O(\log(1/\varepsilon)), \forall y \leq \Theta(\sqrt{\log(1/\varepsilon)})$. Given that $n = \Omega(\frac{d + \log(1/\delta)}{\varepsilon^2})$, with probability $1 - \delta$, it holds that*

$$\|\beta^* - \hat{\beta}\| \leq O(\varepsilon \log(1/\varepsilon))$$

*Proof.* The first order approximate stationary property guarantees

$$\frac{1}{n} \sum_i (y_i - b'(\hat{\beta}^\top \mathbf{x}_i))(\beta^* - \hat{\beta})^\top \mathbf{x}_i \leq \varepsilon \|\beta^* - \hat{\beta}\|$$
$$\implies \frac{1}{n} \sum_{i \in \hat{S} \cap T} (y_i - b'(\hat{\beta}^\top \mathbf{x}_i))(\beta^* - \hat{\beta})^\top \mathbf{x}_i$$
$$\leq -\frac{1}{n} \sum_{i \in \hat{S} \cap E} (y_i - b'(\hat{\beta}^\top \mathbf{x}_i))(\beta^* - \hat{\beta})^\top \mathbf{x}_i + \varepsilon \|\beta^* - \hat{\beta}\|$$

**Lower bound on the LHS**

We first show $y$ is a sub-Gaussian random variable under our assumptions.

**Proposition A.9.** *Suppose a generalized linear model satisfies $b''(\theta) \leq C$ for all $\theta \in \mathbb{R}$, then $(y - \mathbb{E}[y])|\langle \beta^*, \mathbf{x} \rangle$ has sub-Gaussian norm $\sqrt{C}$ for any $\mathbf{x}$*

*Proof.* First note that since a probability density function must sum to one, it hold that for any $\theta \in R$,

$$\int c(y) \exp(\theta y - b(\theta)) dy = 1$$
$$\implies \int c(y) \exp(\theta y) dy = \exp(b(\theta)).$$

Also recall that $\mathbb{E}[y|\theta] = b'(\theta)$. Fix $\theta$, the moment generating function of $y - \mathbb{E}[y]$ can be written as

$$\mathbb{E}[e^{t(y-b'(\theta))}] = \int \exp(t(y - b'(\theta))) \cdot c(y) \exp(\theta \cdot y - b(\theta)) dy$$
$$= \int c(y) \exp((\theta + t)y - b(\theta) - tb'(\theta)) dy$$
$$= \exp(b(\theta + t) - b(\theta) - tb'(\theta)).$$

Note that since $b''(\theta) \le C$, it holds that $b(\theta + t) \le b(\theta) + b'(\theta)t + Ct^2$, and therefore

$$\exp(b(\theta + t) - b(\theta) - tb'(\theta)) \tag{14}$$

$$\le \exp(Ct^2). \tag{15}$$

This implies $y - \mathbb{E}[y]|\langle \beta^*, \mathbf{x} \rangle$ is $\sqrt{C}$-sub-Gaussian random variable. $\qquad\square$

Note that $\mathbb{E}[y|\mathbf{x}] = b'(\langle \beta^*, \mathbf{x} \rangle) \le C\langle \beta^*, \mathbf{x} \rangle + b'(0)$. Since $\langle \beta^*, \mathbf{x} \rangle$ has sub-Gaussian norm 1, $\mathbb{E}[y|\mathbf{x}] - b'(0)$ is a $C$-sub-Gaussian random variable. Therefore $y - b'(0)$ is has sub-Gaussian norm $C + 1$. Therefore, the following resilience property holds just like in the Gaussian case.

**Proposition A.10** (Resilience condition for generalized linear regression). *With probability $1 - \delta$ it holds that for all $Q \subset T$ with $|Q| \ge (1 - 2\varepsilon)n$,*

$$\frac{1}{n}\|\sum_{i \in Q} \left(y_i - b'(\beta^{*\top}\mathbf{x}_i)\right)\mathbf{x}_i\| \le \Theta(\varepsilon \log(1/\varepsilon)).$$

*and for all $Q \subset T$ with $Q \le \varepsilon n$,*

$$\frac{1}{n}\sum_{i \in Q} y_i \le \Theta(\varepsilon\sqrt{\log(1/\varepsilon)})$$

$$\frac{1}{n}\|\sum_{i \in Q} y_i\mathbf{x}_i\| \le \Theta(\varepsilon \log(1/\varepsilon))$$

We conclude that

$$\frac{1}{n}\sum_{i \in \hat{S} \cap T} \left(y_i - b'(\beta^{*\top}\mathbf{x}_i)\right)(\beta^* - \hat{\beta})^\top \mathbf{x}_i \le \varepsilon \log(1/\varepsilon)\|\beta^* - \hat{\beta}\| \tag{16}$$

Next, note that by the lower bound on $b''(\cdot)$, it holds that

$$\sum_{i \in (\hat{S} \cap T)} b''(\beta^\top \mathbf{x}_i)\mathbf{x}_i\mathbf{x}_i^\top \succeq \Theta(I),$$

and hence by strong convexity

$$\sum_{i \in \hat{S} \cap T} \left(b'(\beta^{*\top}\mathbf{x}_i) - b'(\hat{\beta}^\top \mathbf{x}_i)\right)(\beta^* - \hat{\beta})^\top \mathbf{x}_i \ge \Theta(\|\beta^* - \hat{\beta}\|^2)$$

Together we get

$$\sum_{i \in \hat{S} \cap T} (y_i - b'(\hat{\beta}^\top \mathbf{x}_i))(\beta - \hat{\beta})^\top \mathbf{x}_i \ge \Theta(\|\beta^* - \hat{\beta}\|^2) + \Theta(\varepsilon \log(1/\varepsilon)\|\beta^* - \hat{\beta}\|)$$

**Upper bound on the RHS**
**1. Upper bound on the negative log-likelihood.**

Define random variable $z_i = -\log f(y_i|\langle \beta^*, \mathbf{x}_i \rangle)$. Since $y_i - \mathbb{E}[y_i|\mathbf{x}_i]$ is a sub-Gaussian random variable, for a given $\mathbf{x}_i$, $\Pr(y_i - \mathbb{E}[y_i] \ge \sqrt{\log(1/\delta)}) \le \delta$, thus

$$\Pr(z_i \ge \log(1/\delta)) = \Pr(z_i \ge \log(1/\delta)|y_i \le \sqrt{\log(1/\delta)}) + \Pr(y_i \ge \sqrt{\log(1/\delta)})$$

$$\le \delta\sqrt{\log(1/\delta)} + \delta$$

$$\implies \Pr(z_i \ge t) \le e^{-t}\sqrt{t} \le e^{-t/2}.$$

Since this is true regardless of $\mathbf{x}_i$, we have $z_i$ is a sub-exponential random variable. By the resilience property (Corollary D.2), we have that

$$\frac{1}{n}\sum_{i \in T \setminus \hat{S}} -\log f(y_i|\langle \beta^*, \mathbf{x}_i \rangle) = \frac{1}{n}\sum_{i \in T \setminus \hat{S}} z_i \le \Omega(\varepsilon \log(1/\varepsilon)). \tag{17}$$

**2. Turn likelihood bound into square error bound**. Let $\theta = \langle \beta^*, \mathbf{x} \rangle$, and $g(\cdot) = b'^{-1}(\cdot)$. Taking derivative of the negative likelihood function over the mean yields $\frac{b'(\theta) - y}{b''(\theta)}$. This implies for each $y$, $b'(\theta) = y$ minimize the negative log-likelihood function, and

$$(-\log f(y|\theta)) \geq (-\log f(y|g(y))) + (b'(\theta) - y)^2/C.$$

Next we lower bound the likelihood of the minimizer

$$-\log f(y|g(y)) = b(g(y)) - yg(y) - \log(c(y))$$

Since $b(0) = 0$, and note that since $b''(\theta) \leq C$

$$b(0) \leq b(\theta) - b'(\theta)\theta + C\theta^2.$$

This implies

$$-\log f(y|g(y)) \geq -Cg(y)^2 - \log(c(y))$$

Since $b'(x) \geq C_0 x$, it holds that $g(y) \leq y/C_0$. This implies

$$-\log f(y|g(y)) \geq -Cy^2/C_0 - \log(c(y)). \tag{18}$$

Since after truncation $y_i$ are all bounded by $\Theta(\sqrt{\log(1/\varepsilon)})$ from Proposition A.10. We have

$$\frac{1}{n} \sum_{i \in \hat{S} \cap E} -\log f(y_i | \langle \beta^*, \mathbf{x}_i \rangle) \geq \Theta(\frac{1}{n} \sum_{i \in \hat{S} \cap E} (y_i - b'(\beta^{*\top} \mathbf{x}_i))^2) - \Theta(\varepsilon \log(1/\varepsilon)).$$

Combining this with the negative log-likelihood upper bound proved for $T \setminus \hat{S}$, we have

$$\frac{1}{n} \sum_{i \in \hat{S} \cap E} (y_i - b'(\beta^{*\top} \mathbf{x}_i))^2 \leq \Theta(\varepsilon \log(1/\varepsilon))$$

By Cauchy Schwaz

$$-\frac{1}{n} \sum_{i \in \hat{S} \cap E} (y_i - b'(\hat{\beta}^\top \mathbf{x}_i))(\beta^* - \hat{\beta})^\top \mathbf{x}_i \leq \sqrt{\frac{1}{n} \sum_{i \in \hat{S} \cap E} (y_i - b(\hat{\beta}^\top \mathbf{x}_i))^2} \sqrt{\frac{1}{n} \sum_{i \in \hat{S} \cap E} ((\beta^* - \hat{\beta})^\top \mathbf{x}_i)^2}$$

$$\lesssim \varepsilon \log(1/\varepsilon) \|\beta^* - \hat{\beta}\| \tag{19}$$

Combining Equation 19 with the lower bound yields

$$\|\beta^* - \hat{\beta}\| = O(\varepsilon \log(1/\varepsilon)).$$

$\square$

## A.5 Convergence analysis of the alternating minimization algorithm

**Proposition A.11.** *Suppose* $\langle \nabla f(\hat{\beta}), \frac{\beta^* - \hat{\beta}}{\|\beta^* - \hat{\beta}\|} \rangle = \Delta$, *and* $\forall \beta$, $\frac{1}{\|\beta^* - \hat{\beta}\|^2}(\beta^* - \hat{\beta})^\top \nabla^2 f(\beta)(\beta^* - \hat{\beta}) \leq H$. *There exists a point* $\beta$ *such that*

$$f(\beta) \leq f(\hat{\beta}) - \frac{\Delta^2}{4H}.$$

**Lemma A.12** (Algorithm 1 finds an approximate stationary point for generalized linear model)**.** *Given a set of datapoints* $S = \{\mathbf{x}_i, y_i\}_{i=1}^n$ *generated by a generalized linear model with* $\varepsilon_c$-*fraction of corruption. Assuming that* $b''(\mu) \leq C$ *for the generalized linear model and* $n = \Omega(\frac{d + \log(1/\delta)}{\varepsilon^2})$, *then with probability* $1 - \delta$, *the output of Algorithm 1,* $\hat{\beta}$, *is a* $\max(2\sqrt{C\eta}, \frac{2\eta}{\|\beta^* - \hat{\beta}\|})$-*approximate stationary point with input parameters* $R \geq \|\beta^*\|$, $\varepsilon = \varepsilon_c$. *In particular, when* $R = \infty$, *the algorithm returns a* $2\sqrt{C\eta}$-*approximate stationary point.*

*Proof.* Given that Algorithm 1 returns $\hat{\beta}$, and the correponding set is $\hat{S}$. Let $f(\beta)$ be the negative log likelihood function. Then $\nabla^2 f(\beta) = \frac{1}{n}\sum_{i \in \hat{S}} b''(\beta^\top x_i)\mathbf{x}_i\mathbf{x}_i^\top \preceq C \cdot I$. Therefore, $\frac{1}{\|\beta^* - \hat{\beta}\|^2}(\beta^* - \hat{\beta})^\top \nabla^2 f(\beta)(\beta^* - \hat{\beta}) \le C$. Applying Proposition A.11, it holds that there exists $\beta = \frac{\Delta}{2C}\frac{\beta^* - \hat{\beta}}{\|\beta^* - \hat{\beta}\|} + \hat{\beta}$ such that

$$f(\beta) \le f(\hat{\beta}) - \frac{\Delta^2}{4C}.$$

When $R = \infty$, since the algorithm terminate when it is impossible to make an $\eta$ improvement over the current point $\hat{\beta}$, it holds that

$$|\Delta| \le 2\sqrt{C\eta},$$

and $\hat{\beta}$ is a $\sqrt{2C\eta}$-approximate stationary point.

When $R$ only satisfies $\|\beta^*\| \le R$, we need to make sure $\beta$ is a convex combination of $\beta^*$ and $\hat{\beta}$ to make sure it is a valid solution for the optimization problem. Therefore, if $\Delta/2C \ge \|\beta^* - \hat{\beta}\|$, we have

$$f(\beta^*) \le f(\hat{\beta}) - \Delta\|\beta^* - \hat{\beta}\|/2$$

which implies

$$\Delta \le \frac{2\eta}{\|\beta^* - \hat{\beta}\|}$$

$\square$

Since $b''(\theta) = \exp(\theta)$ is unbounded for Poisson regression, the above analysis does not apply, and here we prove our alternating minimization algorithm still returns an approximate stationary point

**Lemma A.13** (Algorithm 1 finds an approximate stationary point for Poisson regression (Restatement of Lemma 4.8)). *Given a set of datapoints $S = \{\mathbf{x}_i, y_i\}_{i=1}^n$ generated by a Poisson model with $\varepsilon_c$-fraction of corruption. Assuming that $n = \Omega(\frac{d + \log(1/\delta)}{\varepsilon^2})$, then with probability $1 - \delta$, the output of Algorithm 1 with input parameters $\varepsilon = 2\varepsilon_c, R \ge \|\beta^*\|, \eta = \varepsilon^2/(dn)$, is a $\max(\varepsilon, \frac{2\varepsilon^2}{\|\beta^* - \hat{\beta}\|})$-approximate stationary point.*

*Proof.* Define

$$H = \frac{1}{n\|\beta^* - \hat{\beta}\|^2}\sum_{i \in \hat{S}}\exp(\hat{\beta}^\top \mathbf{x}_i)((\beta^* - \hat{\beta})^\top \mathbf{x}_i)^2,$$

to be the second order derivative along the $\beta^* - \hat{\beta}$ direction. For every point $\beta = (1 - \lambda)\hat{\beta} + \lambda\beta^*, 0 \le \lambda \le 1$, the second order derivative is

$$\frac{1}{n\|\beta^* - \hat{\beta}\|^2}\sum_{i \in \hat{S}}\exp(((1 - \lambda)\hat{\beta} + \lambda\beta^*)^\top \mathbf{x}_i)((\beta^* - \hat{\beta})^\top \mathbf{x}_i)^2$$

$$\le \frac{1}{n\|\beta^* - \hat{\beta}\|^2}\left(\sum_{i \in \hat{S}}\exp(\hat{\beta}^\top \mathbf{x}_i)((\hat{\beta} - \beta^*)^\top \mathbf{x}_i)^2 + \sum_{i \in \hat{S}}\exp(\beta^{*\top}\mathbf{x}_i)((\beta^* - \hat{\beta})^\top \mathbf{x}_i)^2\right)$$

$$\le H + \frac{1}{n\|\beta^* - \hat{\beta}\|^2}\sum_{i \in \hat{S}}\exp(\beta^{*\top}\mathbf{x}_i)((\beta^* - \hat{\beta})^\top \mathbf{x}_i)^2$$

Since $\beta^{*\top}\mathbf{x}_i$ is sub-Gaussian, it holds that $\max_{i \in \hat{S}}\beta^{*\top}\mathbf{x}_i \le \sqrt{\log(n)}$ with probability 0.99. Hence with probability 0.99, we have

$$\le H + \exp(\sqrt{\log(n)})$$

Consider two scenarios for the value of $H$.
1. $H \le C \cdot dn$.

We can apply the same argument as in Lemma A.12 and obtain that $\hat{\beta}$ is a $\max(2\sqrt{dn\eta}, \frac{2\eta}{\|\beta^* - \hat{\beta}\|})$ approximate stationary point. Plugging in that $\eta = \varepsilon^2/(dn)$, we get that $\hat{\beta}$ is a $\max(\varepsilon, \frac{\varepsilon^2}{\|\beta^* - \hat{\beta}\|})$ approximate stationary point.

*2. $H \geq C \cdot dn$.*

We will show this can not happen due to the termination condition of our algorithm. Due to the norm concentration of sub-Gaussian random vector, with probability 0.99, it holds that $\max_{i \in \hat{S}} \|\mathbf{x}_i\|^2 \leq d + \Theta(\sqrt{d})$. Denote $D = \sum_i \exp(\hat{\beta}^\top \mathbf{x}_i)$. Then

$$D = \frac{1}{n} \sum_{i \in \hat{S}} \exp(\hat{\beta}^\top \mathbf{x}_i) \geq H / \max_{i \in \hat{S}} \|\mathbf{x}_i\|^2 \geq \Theta(H/d).$$

The following inequality always holds (log sum inequality)

$$\frac{1}{n} \sum_{i \in \hat{S}} \exp(\hat{\beta}^\top \mathbf{x}_i)(\hat{\beta}^\top \mathbf{x}_i) \geq D \log(D).$$

The following inequality holds since $\beta^{*\top} \mathbf{x}_i \leq \sqrt{\log(n)}$

$$\frac{1}{n} \sum_i \exp(\hat{\beta}^\top \mathbf{x}_i)(\beta^{*\top} \mathbf{x}_i) \leq D \sqrt{\log(n)}$$

Therefore,

$$\frac{1}{n} \sum_i \exp(\hat{\beta}^\top \mathbf{x}_i)(\hat{\beta}^\top - \beta^*)^\top \mathbf{x}_i \geq D(\log(D) - \sqrt{\log n}) \geq \frac{H}{\Theta(d)}(\log(H/\Theta(d)) - \sqrt{\log n})$$

Since $H \geq \Theta(dn)$, it holds that

$$\Delta := \frac{1}{n} \sum_i \exp(\hat{\beta}^\top \mathbf{x}_i)(\hat{\beta}^\top - \beta^*)^\top \mathbf{x}_i \geq \Theta(\frac{H \sqrt{\log n}}{d})$$

and that the gradient satisfies of the

$$\frac{1}{n} \sum_{i \in \hat{S}} (\exp(\hat{\beta}^\top \mathbf{x}_i) - y_i)(\hat{\beta}^\top - \beta^*)^\top \mathbf{x}_i$$

$$\geq \frac{H}{d} - \exp(\sqrt{\log(1/\varepsilon)})\sqrt{d}$$

$$\geq \Theta(H/d),$$

where the first inequality holds since $y_i \leq \exp(\sqrt{\log(1/\varepsilon)})$, $\|\mathbf{x}_i\| = \Theta(\sqrt{d})$, and the second inequality holds since $n \geq 2\sqrt{d}/\varepsilon \geq 2\exp(\sqrt{\log(1/\varepsilon)})\sqrt{d}$.

Note that in this regime the second order derivative along the $\beta^* - \hat{\beta}$ direction is upper bounded by $2H$ for every point $\beta = (1-\lambda)\hat{\beta} + \lambda\beta^*, 0 \leq \lambda \leq 1$. Therefore, for every point in $\beta_\lambda = (1-\lambda)\hat{\beta} + \lambda\beta^*$, the value of the likelihood function is bounded as

$$-\frac{1}{n} \sum_{i \in \hat{S}} \log f(y_i|\langle \beta, \mathbf{x}_i \rangle) \leq -\frac{1}{n} \sum_{i \in \hat{S}} \log f(y_i|\langle \hat{\beta}, \mathbf{x}_i \rangle) - \lambda\Delta + H\lambda^2$$

If $\Delta \leq 2H$, setting $\lambda = \Delta/2H$ yield

$$-\frac{1}{n} \sum_{i \in \hat{S}} \log f(y_i|\langle \beta, \mathbf{x}_i \rangle) \leq -\frac{1}{n} \sum_{i \in \hat{S}} \log f(y_i|\langle \hat{\beta}, \mathbf{x}_i \rangle) - \Delta^2/4H.$$

Since $\Delta \gtrsim H/d$, and $H \geq nd$, the drop in objective value $\Delta^2/4H = \Omega(n/d) \geq \Omega(1/\varepsilon^2)$.

If $\Delta \geq 2H$, setting $\lambda = 1$ yield

$$-\frac{1}{n} \sum_{i \in \hat{S}} \log f(y_i|\langle \beta, \mathbf{x}_i \rangle) \leq -\frac{1}{n} \sum_{i \in \hat{S}} \log f(y_i|\langle \hat{\beta}, \mathbf{x}_i \rangle) - H,$$

and the drop in objective value is $H \geq nd >> 1/\varepsilon^2$. Since $\eta \leq 1/\varepsilon^2$, we conclude that $H$ must be smaller than $C \cdot dn$ when the algorithm terminate.

$\square$

# B Handling sample corruption model

This section we discuss the algorithm for the sample corruption model, where the adversary is allowed to change the covariate $\mathbf{x}_i$ in addition to label $y_i$. At the end of Algorithm 4 in [DHL19], with high probability, it will return a set of $\varepsilon$ corrupted data points that satisfies $\|\frac{1}{n}\sum_{i\in\hat{S}}\mathbf{x}_i\mathbf{x}_i^\top - I\| = O(\varepsilon\log(\varepsilon))$. Corollary D.3 then implies that $\|\frac{1}{n}\sum_{i\in\hat{S}\cap E}\mathbf{x}_i\mathbf{x}_i^\top\| = O(\varepsilon\log(\varepsilon))$. The rest of the proof proceeds as in the label corruption setting.

# C Non-identity covariance

Our result of Theorem 4.2, Theorem 4.3, Theorem 4.4 apply to the case where the covariance of $\mathbf{x}_i$ is identity. Here we argue that the result holds for general covariance $\Sigma$ where the guarantee is in terms of $\|\hat{\beta} - \beta^*\|_\Sigma$, which is the root-mean-square error in the Gaussian setting. First note that Algorithm 1 on input $(S = (\mathbf{x}_1, y_1), \ldots, (\mathbf{x}_n, y_n), \varepsilon, \eta, R)$ with non-identity covariance $\Sigma$ output $\hat{\beta}$ if and only if running it on input $S = (\Sigma^{-1/2}\mathbf{x}_1, y_1), \ldots, (\Sigma^{-1/2}\mathbf{x}_n, y_n), \varepsilon, \eta$ with the constraint $\|\Sigma^{-1/2}\beta\|_2 \leq R$ output $\Sigma^{1/2}\hat{\beta}$.

Although the constraint $\|\Sigma^{-1/2}\beta\|_2 \leq R$ is different from the $\ell_2$ constraint in the analysis of Algorithm 1, with slightly different arguments as in Theorem 4.2, Theorem 4.3, Theorem 4.4, we can show that running Algorithm 1 on $S = (\Sigma^{-1/2}\mathbf{x}_1, y_1), \ldots, (\Sigma^{-1/2}\mathbf{x}_n, y_n), \varepsilon, \eta$ with the constraint $\|\Sigma^{-1/2}\beta\|_2 \leq R$ will output $\hat{\beta}'$ such that $\|\hat{\beta}' - \Sigma^{1/2}\beta\|_2$ has the desired error rate. This implies Algorithm 1 running on $(S = (\mathbf{x}_1, y_1), \ldots, (\mathbf{x}_n, y_n), \varepsilon, \eta, R)$ will output $\hat{\beta} = \Sigma^{-1/2}\hat{\beta}'$ such that $\|\hat{\beta} - \beta\|_\Sigma$ has the desired error rate.

# D Auxiliary Lemmas

**Proposition D.1** (Resilience of sub-Gaussian sample [JLT20, Corollary 4]). *Let $G = \{\mathbf{x}_i \in \mathbb{R}^d\}_{i=1}^n$ be a dataset satisfies Assumption 3.3. For any $\varepsilon \in [0, 1/2]$. If*

$$n = \Omega\left(\frac{d + \log(1/\delta)}{\varepsilon^2}\right), \tag{20}$$

*then with probability at least $1 - \delta$ there exists an absolute constant $C > 0$ such that for any subset $T \subset G$ and $|T| \geq (1 - \varepsilon)n$, we have*

$$\|\frac{1}{|T|}\sum_{i\in T}\mathbf{x}_i\| \leq C\varepsilon\sqrt{\log(1/\varepsilon)} \tag{21}$$

$$\|\left(\frac{1}{|T|}\sum_{i\in T}\mathbf{x}_i\mathbf{x}_i^\top\right) - I\| \leq C\varepsilon\log(1/\varepsilon) \tag{22}$$

*and that for any subset $T \subset G$ and $|T| \geq \varepsilon n$, we have*

$$\|\frac{1}{|T|}\sum_{i\in T}\mathbf{x}_i\| \leq C\sqrt{\log(1/\varepsilon)} \tag{23}$$

$$\|\left(\frac{1}{|T|}\sum_{i\in T}\mathbf{x}_i\mathbf{x}_i^\top\right) - I\| \leq C\log(1/\varepsilon) \tag{24}$$

**Corollary D.2** (Resilience of 1-d sub-exponential sample). *Let $G = \{x_i \in \mathbb{R}^d\}_{i=1}^n$ contains i.i.d. samples drawn from 1 sub-exponential distribution with zero mean. For any $\varepsilon \in [0, 1/2]$. If*

$$n = \Omega\left(\frac{\log(1/\delta)}{\varepsilon^2}\right), \tag{25}$$

*then with probability at least $1 - \delta$ there exists an absolute constant $C > 0$ such that for any subset $T \subset G$ and $|T| \geq \varepsilon n$, we have*

$$|\frac{1}{|T|}\sum_{i\in T}\mathbf{x}_i| \leq C\log(1/\varepsilon) \tag{26}$$

**Corollary D.3** (Resilience of corrupted sample with small covariance). *Let $G = \{\mathbf{x}_i \in \mathbb{R}^d\}_{i=1}^n$ be a dataset satisfies the condition in Proposition D.1 with $\Sigma = I$ and parameter $\varepsilon$. Let $S = G \setminus L \cup E = T \cup E$ be obtained from set $G$ by corrupting $\varepsilon n$ samples arbitrarily. There exists an absolute constant $C > 0$ such that if*

$$\|\frac{1}{n}\sum_{i \in \hat{S}} \mathbf{x}_i \mathbf{x}_i^\top - I\| = O(\varepsilon \log(\varepsilon))$$

*for $\hat{S}$ with $\hat{S} \geq (1-\varepsilon)n$, we have*

$$\frac{1}{n}\|\sum_{i \in \hat{S} \cap T} \mathbf{x}_i\| \leq C\varepsilon\sqrt{\log(1/\varepsilon)}$$

$$\|\left(\frac{1}{n}\sum_{i \in \hat{S} \cap T} \mathbf{x}_i \mathbf{x}_i^\top\right) - I\| \leq C\varepsilon \log(1/\varepsilon)$$

*and*

$$\frac{1}{n}\|\sum_{i \in \hat{S} \cap E} \mathbf{x}_i\| \leq C\varepsilon\sqrt{\log(1/\varepsilon)}$$

$$\|\left(\frac{1}{n}\sum_{i \in \hat{S} \cap E} \mathbf{x}_i \mathbf{x}_i^\top\right)\| \leq C\varepsilon \log(1/\varepsilon)$$

*Proof.* The statement about $T$ follows directly from resilience of sub-Gaussian sample. The statement about $E$ follows from the covariance bound and the condition of $T$. $\square$

**Definition D.4** (Approximate stationary point). *Given a set of datapoints $\{\mathbf{x}_i, y_i\}_{i=1}^n$, probability density function $f(\cdot|\cdot)$, and $\beta^* \in \mathbb{R}^d$, $\hat{\beta} \in \mathbb{R}^d$. Let $\hat{S} = \arg\min_{S:|S|=(1-\varepsilon)n} \sum_{i \in S} -\log f(y_i|\langle\hat{\beta}, \mathbf{x}_i\rangle)$. We call $\hat{\beta}$ a $\eta$-stationary point if*

$$\left(\nabla_\beta \sum_{i \in \hat{S}} -\log f(y_i|\langle\hat{\beta}, \mathbf{x}_i\rangle)\right)^\top \frac{(\hat{\beta} - \beta^*)}{\|\hat{\beta} - \beta^*\|} \leq \eta$$

**Fact D.5** ($k$-th moment bound of Poisson [Ahl22]). *Assuming that $y \sim Poi(\lambda)$, then $\mathbb{E}[y^k] = \max(k^k, \lambda^k)$ and $\mathbb{E}[(y-\lambda)^k] = \max(k^k, \lambda^k)$*

**Fact D.6.** *For all $n > 0$, $n \log n - n + 1 \leq \log n! \leq (n+1)\log n - n + 1$*

**Fact D.7.**

$$\log\binom{m}{y} \leq -y\log(y/m) - (m-y)\log((m-y)/m) - \log\frac{y(m-y)}{m} + C_2$$