# OpenReview forum: "Trimmed Maximum Likelihood Estimation for Robust Generalized Linear Model"
_NeurIPS.cc/2022/Conference — NeurIPS 2022 Accept_

### Official Review · Reviewer_B8AC · 2022-07-10

**Rating:** 7
**Confidence:** 2
**Soundness:** 3 good
**Presentation:** 3 good
**Contribution:** 4 excellent

**Summary:**

This paper studies the theoretical properties of alternating minimization of trimmed log-likelihood loss for generalized linear models with corruption in the label space and joint space.

**Questions:**

The authors have remarked several limitations including sub-Gaussian covariates and a very limited assumption on the covariance for sample corruption model. On top of that, I have a couple of concerns.

1. My major concern is whether the theoretical results are suspicious to be simple corollaries of existing results. First, the authors have appreciated the works [1][2] based on which a majority of their results are developed. The only difference seems to be the adopted loss, which is the negative log-likelihood, and the more general distribution, the generalized linear models. The formulation is reduced to [1] when Gaussian regression is assumed. A similar algorithm is used in [3] but not cited/discussed. I notice that Theorem 1.8 in [4] probably presents a stronger result with milder assumptions than that developed in this paper. Furthermore, Example 4 in [5] and Theorem 1.4 in [6] seem relevant but absent in references. It would be nice if a clear connection can be drawn between this work and [3][4][5][6].

2. The theoretical results may not be fundamental enough to exclude empirical results. The work could've taken advantage of the remaining 0.8 pages to demonstrate some experimental results even with a toy simulation dataset.

3. Line 1 in Algorithm 1 filters out data points with large labels in terms of $\ell_1$ norm. Does it still apply to nominal distributions? Some explanations or references will be appreciated.

4. Although the paper is well-written, there are many issues in terms of grammar, tense and punctuation if one takes a closer look at it. I list a few of them found as follows.
- Line 12, should use bold letters $\boldsymbol{\beta}$ for vectors for consistency.
- Line 35, 'achieve' -> 'achieves'.
- Line 52, 'a' -> 'an'.
- Line 53-55, duplicate 'achieve(s)'; tenses do not match for 'improved' and 'matches'.
- Line 70, 'Section B' -> 'Section 5'.
- Definition 3.1, scalar $\phi$ defined but unused.
- Definition 3.3, comma after the statement.
- A lot of equations and paragraphs end without punctuation.
- Line 129, 'though' -> 'through'.
- Line 144, 'maximize' -> 'maximizes'; 'terminate' -> 'terminates'; 'output' -> 'outputs'.
- Line 183, 'upper bound terms of' -> 'upper bound in terms of'.
- Line 201, period at the end.
- Line 241, duplicate 'be/been applied'.
- Line 247, 'extends' -> 'extended'.


References:

[1] Shen, Yanyao, and Sujay Sanghavi. "Learning with bad training data via iterative trimmed loss minimization." In International Conference on Machine Learning, pp. 5739-5748. PMLR, 2019.

[2] Chen, Sitan, Frederic Koehler, Ankur Moitra, and Morris Yau. "Online and distribution-free robustness: Regression and contextual bandits with huber contamination." In 2021 IEEE 62nd Annual Symposium on Foundations of Computer Science (FOCS), pp. 684-695. IEEE, 2022.

[3] Shen, Yanyao, and Sujay Sanghavi. "Iterative least trimmed squares for mixed linear regression." Advances in Neural Information Processing Systems 32 (2019).

[4] Diakonikolas, Ilias, Daniel M. Kane, Alistair Stewart, and Yuxin Sun. "Outlier-robust learning of ising models under dobrushin’s condition." In Conference on Learning Theory, pp. 1645-1682. PMLR, 2021.

[5] Foster, Dylan J., and Akshay Krishnamurthy. "Efficient first-order contextual bandits: Prediction, allocation, and triangular discrimination." Advances in Neural Information Processing Systems 34 (2021): 18907-18919.

[6] Chen, Sitan, Frederic Koehler, Ankur Moitra, and Morris Yau. "Classification under misspecification: Halfspaces, generalized linear models, and evolvability." Advances in Neural Information Processing Systems 33 (2020): 8391-8403.

**Limitations:**

The authors have partially addressed the limitations. Potential negative societal impact is not mentioned but unnecessary due to the theoretical nature of the work. The studied problem is interesting and results have rigorous technical proofs. If the authors can put this work  in a more clear position in the literature, I believe the paper can improve significantly.

**Strengths And Weaknesses:**

Strengths:

The studied setting, as a combination of the adopted estimator, the distributional assumption and the contamination models, is novel with respect to the cited literature in the manuscript. The asserted results and technical proofs seem solid. The paper is well-written and very easy to follow.

Weaknesses:

Although the setting is novel, the claims in this paper may be trivial extensions of existing results. No empirical results are given. There are a noteworthy number of typos and grammatical issues that render it not publication-ready.

---

> ### Author Response · Authors · 2022-08-02
> **Response to reviewer B8AC**
>
> Thanks so much for your thoughtful review and questions. Please find responses below.
>
> **Q: “First, the authors have appreciated the works [1][2] based on which a majority of their results are developed. The only difference seems to be the adopted loss, which is the negative log-likelihood, and the more general distribution, the generalized linear models. The formulation is reduced to [1] when Gaussian regression is assumed.”**
>
> A: We would like to emphasize that iterative trimmed loss estimator is a generic approach for robust learning that dates back to Lagrange and is widely used in practice. While [1] also used iterative trimmed loss estimators, the error rate they obtained is suboptimal since it stays as O(\sigma) regardless of the corruption level, on the other hand, our analysis is very different from it and showed a nearly optimal rate. On a very high level, the framework of our analysis is similar to [2], but the details are very different since they worked on the distribution-free setting with squared loss. Finally, generalizing our result from squared loss to other negative log-likelihood loss is highly non-trivial. For example in Poisson regression, since the mean function is an exponential function, the negative log-likelihood of even a good example can be exponentially large which will ruin our error bound. To get around this, we need to rely on the resilience property to argue that for a good set of data points of size eps*n, there must be at least one data point with small negative log-likelihood, which then implies all the data points selected by the algorithm have small negative log-likelihood.
>
> **Q: “Theorem 1.8 in [4] probably presents a stronger result with milder assumptions than that developed in this paper”**
>
> A: From our understanding, Theorem 1.8 in [4] concerns robustly estimating the parameters of a distribution from the exponential family, which is very different from our regression problem. It is not obvious to us how it will imply a stronger result in the setting of robustly learning GLM.
>
> **Q: “Theorem 1.4 in [6] seem relevant but absent in references”**
>
> A: Theorem 1.4 in [6] considered misspecified GLMs with binary response variable only, while our result applies to much more general GLMs including Gaussian, Poisson and Binomials. The misspecification setting considered in their paper is also incomparable to our setting, roughly speaking, they allow mild semi-random corruption on a large number of examples, while our model allows adversarial corruption on a small fraction of examples. We have cited the paper as a related work on robust classification/logistic regression.
>
> **Q: “Example 4 in [5] seem relevant but absent in references”**
>
> A: Example 4 in [5] is about kernelized contextual bandits which seems unrelated to our work. Maybe the reviewer is referring to Example 5 in [5] which concerns regret bound for online logistic regression? The setting is quite different from ours, in particular there is no corruption, and it is not clear to us how that work implies results for learning GLM under adversarial corruption.
>
> **Q: “A similar algorithm is used in [3] but not cited/discussed”**
>
> A: It is true that [3] also used trimmed least squares estimators as in our paper. However, they considered robust learning of mixed linear regression with no random noise on the label which is incomparable to our result. We have added [3] as a related work on trimmed least squares estimators.
>
> **Q: “The theoretical results may not be fundamental enough to exclude empirical results.”**
>
> A: The idea of iterative trimmed loss estimation is folklore that dates back to Lagrange, and the contribution of our work is not proposing this estimator, but theoretically proving the near optimality of this heuristic approach. The practical effectiveness of the iterative trimmed loss estimator has been demonstrated in previous works. For example, [1] showed its robustness against bad training data in both linear regression and image classification on MNIST and CIFAR-10.
>
> **Q: “Line 1 in Algorithm 1 filters out data points with large labels in terms of l1 norm. Does it still apply to nominal distributions?”**
>
> A: We could not figure out the definition of “nominal distribution”. Is it supposed to be “normal distributions”? For normal distribution, line 1 of Algorithm 1 will not be necessary. Although, as we showed, the guarantee of the algorithm stays the same regardless of the filtering step in line 1.
>
> Thanks for pointing out the typos! We have fixed all the typos in the revised version.

---

> > ### Comment · Reviewer_B8AC · 2022-08-03
> > **A few more questions**
> >
> > Thanks for the detailed clarifications which have addressed most of my concerns. Now I am able to appreciate the fundamental contributions of this paper. The revised related works now looks much better to me. Although resilience seems like a strong assumption as pointed out by other reviewers, it is probably the best we can do now. I have one more question as follows.
> >
> > Line 1 in Algorithm 1 is optional for theoretical results. In practice, it yields good initialization. Does filtering out points with large label norms depend on the zero-mean assumption of x in Definition 3.3? If x has a non-zero mean, is Line 1 still applicable?
> >
> > By the way, Online and Distribution-Free Robustness: Regression and Contextual Bandits with Huber Contamination has been accepted to FOCS so please revise the reference entry accordingly.

---

> > > ### Author Response · Authors · 2022-08-05
> > > **Response to reviewer B8AC**
> > >
> > > Thank you for your questions. Please find responses below.
> > >
> > > **Q: Line 1 in Algorithm 1 is optional for theoretical results. In practice, it yields good initialization. Does filtering out points with large label norms depend on the zero-mean assumption of x in Definition 3.3? If x has a non-zero mean, is Line 1 still applicable?**
> > >
> > > A: Due to the resilience of the sample, removing any epsilon fraction of the datapoints won’t change the resilience property of the dataset which is needed for our proof. Therefore, Line 1 can never cause a problem. The main purpose of Line 1 is to limit the range of y, which is needed for the proof of Poisson and general GLMs. When x has non-zero mean, the rate of Gaussian regression will not change with or without Line 1.
> > >
> > > We have updated the citation of “Online and Distribution-Free Robustness: Regression and Contextual Bandits with Huber Contamination” accordingly.
> > >
> > > We hope we have adequately addressed all the concerns. If so, we would be grateful if the reviewer can kindly reflect that in the score. We are here to answer any further questions.

---

> > > > ### Comment · Reviewer_B8AC · 2022-08-06
> > > > **Rating updated**
> > > >
> > > > Thanks for clarifying. I have updated my rating to vote for accept.

---

### Official Review · Reviewer_nLxp · 2022-07-11

**Rating:** 7
**Confidence:** 4
**Soundness:** 3 good
**Presentation:** 1 poor
**Contribution:** 3 good

**Summary:**

The paper studies generalized linear models when a fraction of data is corrupted by an adversary. The main focus is on analyzing Trimmed Maximum Likelihood Estimator(TML), a popular estimator for this setting. TML is defined as a solution to a non-convex optimization problem and the standard off-the-shelf guarantees for convex optimization do not apply. The main contribution of the paper is to develop strong finite-time guarantees for the statistical properties of this estimator for various link functions. In particular, the results are significantly tighter than that of Bhatia et al. 2015 and 2017. The paper begins by analyzing the setting where the covariates are clean and only the labels are corrupted. Then the results are extended to the setting where the covariates may also have been corrupted.

**Questions:**

+ See above.
+ The proofs in Appendix are not very clear. More critically, there are questions regarding soundness. For example, in Line 395, why is the expression upper bounded by O(1)? Should it not depend on R, which can be \infty in this setting?

**Limitations:**

The paper should add potential limitations of their results.

**Strengths And Weaknesses:**


## Strengths
The main contribution is to show any approximate stationary point of TML, which can be obtained by an alternating minimization algorithm, has desirable statistical properties. This result is technically interesting and practically relevant. Thus, it would be of interest to the NeurIPS community.

## Preliminary Recommendation
I like the paper, and my preliminary recommendation is "Accept".  I look forward to reading the authors' response addressing the points below.

## Weaknesses
However, the paper needs significant improvements on several fronts mentioned below, and I look forward to reading the authors' response addressing these points.

1. Related work: I feel the literature review of the current paper should include the following:

	+ (Covariate Filtering approach of [PJL20]) The paper's approach in Section 5 to handle corruptions in covariates is very similar to [PJL20]. Please include a detailed comparison. In particular, the lines 36 and 241 need to be modified.

	+ Other works in robust statistics (and ML in general) show that approximate stationary points of a non-convex problem have good statistical properties, for example, [CDGS20, CDKGGS21, ZJS20].


2. Technically imprecise statements: I feel the writing is sloppy at times. For example,
	+ Line 35: trimmed MLE estimator does not achieve the minimax error rate (Error is off by a logarithmic factor).
	+ In the current form, the writing in Line 27 gives the impression that [33] is minmax optimal when the noises are heavy-tailed. I am not sure if that is the case.
	+ Line 130: It is not readily obvious to me that the results adapt to the unknown covariance setting in the label corruption model; I recommend adding explicit details and precise proofs.
	+ Line 96: I believe the results in [PJL20] are tighter for the Gaussian regression.

	+ I recommend adding explicit dependence on the norm of R in the error guarantees instead of hiding it in the big-O.



3. Writing:
	+ I liked that the paper gives a brief proof sketch of the proof of stationary points being statistically efficient. However, the proof sketch is unclear in several places. For example, "resilience" is not defined in the main body (the paper can use the additional space available on the 9th page). I strongly suggest improving the prose in this proof sketch in the next version because it is the main technical contribution.

     + (Appendix) The Appendix in particular needs to be heavily revised. Some specific comments are below:
         -    The proofs in Appendix are written in a very unorganized manner. It is not obvious from the beginning what statistical conditions are required from the inliers (see also the remark in the Questions section below). It is thus difficult to verify the soundness of the proofs (since there is an adaptive adversary involved, one is prone to to incorrectly claim that certain random variables are independent). I recommend reorganizing and rewriting the proofs in a coherent manner.
         -  The proofs in Appendix are not described clearly. Majority of the text in the proofs do not give a high-level intuition of the proof strategy. For example, they are along the lines of "Observe that", "it holds that", and so on. See also the formatting on Page 15.


   - Furthermore, there are numerous typos in the paper.
     + Line 78: "proposed"
	  + Line 144: "terminates"
	 + Line 191: "mmoment"
	 + Line 201: missing period
	 + Line 206: "intro", Lemma 4.5
	 + Lemma 4.5, Lemma 4.6: Given "as".
	 + Line 247: "extends"
	 + Definition 3.1:  "exists" -> exist. Where is scalar $\phi$?
	 + Display after Line 162: There should be a parenthesis enclosing the expression before \top.

### Update
I had forgotten to include the detailed bibliography for the references:

[PJL20] A. Pensia, V. Jog, P. Loh. 2020. Robust regression with covariate filtering: Heavy tails and adversarial contamination

[CDGS20] Y. Cheng, I. Diakonikolas, R. Ge, M. Soltanolkotabi. 2020. High-Dimensional Robust Mean Estimation via Gradient Descent

[CDKGGS21] Y. Cheng, I. Diakonikolas, R. Ge, S. Gupta, D. Kane, M. Soltanolkotabi. 2021. Outlier-Robust Sparse Estimation via Non-Convex Optimization

[ZJS20] B. Zhu, J. Jiao, J. Steinhardt. 2020. Robust estimation via generalized quasi-gradients

---

> ### Author Response · Authors · 2022-08-02
> **Response to reviewer nLxp**
>
> Thanks for pointing out the fact that the rate of trimmed MLE estimators is actually sub-optimal by a sqrt log factor, the connection to PJL20, and related works on approximate stationary points. We have modified related statements accordingly.
>
> **Q: “not readily obvious the results adapt to the unknown covariance setting in the label corruption model”**
>
> A: Suppose the true covariance of $x$ is $\Sigma$. Given a first order stationary point $\hat{\beta}$ of a set of data points $\{x_1,...x_n\}$, we can argue that $\Sigma^{1/2} \hat{\beta}$ is a first order stationary point of whitened $x$, i.e. $\{\Sigma^{-1/2}x_1,..., \Sigma^{-1/2}x_n\}$. Our proof implies that $\Sigma^{1/2}\hat{\beta}$ is close to $\Sigma^{1/2}{\beta}$, and this will guarantee $\hat{\beta}$ approximate $\beta$. We agree this is not obvious and we will add explicit details and precise proofs.
>
> **Q: “Line 395, why is the expression upper bounded by $O(1)$? Should it not depend on $R$, which can be $\infty$ in this setting?”**
>
> A: You are correct that the expression is about $||\beta^*||^2 \le R^2$. However, as stated in line 128, we assume $||\beta^*||^2$ is a constant, which is why the quantity is $O(1)$.
>
> Thanks for the writing comments! All the typos are fixed in the revised version. We will incorporate the general writing comments in the final version.

---

> > ### Comment · Reviewer_nLxp · 2022-08-03
> > **Thank you for the repsonse**
> >
> > I thank the authors for their thoughtful response and including the suggested changes. I am sorry I forgot to add a bibliography at the end of my review; I have updated the review now. I thank the authors for their effort in still finding these references.
> >
> > However, I still think that a few of my concerns have not been answered.
> >
> > 1. **Comparison with [PJL20] and other prior works**: I still believe the work of [PJL20] ([33] in the new version) is not properly cited and discussed.
> >
> >    + Line 102 currently says that "*[33] extended the iterative thresholding algorithm to the heavy-tailed and corrupted covariate settings and obtained better rate while we leveraged the same approach to simultaneously handle covariate and label corruptions*".
> >
> >       They also studied corruptions in both labels and covariates (not just Gaussian, but also heavy-tailed). Line 102 particularly seems to *oversell* the current paper's contribution. This is surprising because
> >
> >    + Algorithm 3 (in the Appendix) of the present paper is **identical** to the algorithm in [PJL20].
> >
> >        Given such a strong connection, I would expect a more detailed and fair discussion on their work up front.  In particular, what are the contributions of the present work if the algorithm is the same and the proof strategy is also similar (the analysis of [PJL20] also relies on resilience, which they call "stability")?
> >
> >    + Line 98: The paper says that the best dependence on $\epsilon$ in the prior works of [4,5] was $O(\sigma)$ without any dependence on $\epsilon$.
> >
> >        However, [PJL20, Lemma 4.1] has mentioned that the dependence on $\epsilon$ in the prior work of [4] is in fact $\sqrt{\epsilon}$.  This seems to be another instance where the prior work is not properly cited.
> >
> > 2. **(Dependence on the norm bound $R$)** I would like to see the explicit dependence on $R$. Can the authors please comment on the dependence on $R$ in the runtime, sample complexity, and the error rate for the GLMs considered in the paper? In many settings, the dependence on $R$ turns out to be crucial, e.g., logistic regression.
> >
> > 3. (**Proof sketch in the main body**) It is my understanding that the proof sketch has not changed in the new version. Please let me know if that is correct. As I said in my review, the proof sketch needs to be clearer.
> >
> > I look forward to reading the authors' response.

---

> > > ### Author Response · Authors · 2022-08-06
> > > **Response to reviewer nLxp**
> > >
> > > Thank you for your questions. Please find responses below.
> > >
> > > **Q: Line 102 currently says that "[33] extended the iterative thresholding algorithm to the heavy-tailed and corrupted covariate settings and obtained better rate while we leveraged the same approach to simultaneously handle covariate and label corruptions". They also studied corruptions in both labels and covariates (not just Gaussian, but also heavy-tailed). Line 102 particularly seems to oversell the current paper's contribution.**
> > >
> > > A:  We are sorry for the misunderstanding in the language of the previous version. To avoid any confusion, we have modified the statement and it now reads “[PJL20] extended the iterative thresholding algorithm to the heavy-tailed covariate setting and can simultaneously handle both labels and covariate corruptions”.
> > >
> > > **Q: Algorithm 3 (in the Appendix) of the present paper is identical to the algorithm in [PJL20].
> > > Given such a strong connection, I would expect a more detailed and fair discussion on their work up front. In particular, what are the contributions of the present work if the algorithm is the same and the proof strategy is also similar (the analysis of [PJL20] also relies on resilience, which they call "stability")?**
> > >
> > > A: Under the Gaussian regression setting, our Algorithm 3 is identical to Algorithm 3 of [PJL20] . However, our contribution over the analysis of Algorithm 3 in [PJL20] is the following:
> > > 1 .We obtain an improved error rate in the Gaussian regression setting. In particular, Lemma 4.1 in [PJL20] implies a $O(\sigma\sqrt{\epsilon})$ error bound while we proved a $O(\sigma \epsilon \log(1/\epsilon)$ error bound.
> > > 2. Algorithm 3 of [PJL20] does not consider other GLMs, while our result applies to GLMs including Poisson and Binomial regression.
> > > We have added a paragraph in the related work section to compare to [PJL20]
> > >
> > > **Q: Line 98: The paper says that the best dependence on $\epsilon$ in the prior works of [4,5] was $O(\sigma)$  without any dependence on $\epsilon$. However, [PJL20, Lemma 4.1] has mentioned that the dependence on $\epsilon$  in the prior work of [4] is in fact $\sqrt{\epsilon}\sigma$. This seems to be another instance where the prior work is not properly cited.**
> > >
> > > A: In line 98, we only referred to [5] and not [4]. We did not mention the result in [4] here since it considers an oblivious corruption setting which is different from our adversarial setting, and the estimator in [4] can even achieve a consistent estimate which is impossible in the adversarial setting. Thanks for pointing out that [PJL20, Lemma 4.1] mentioned that by adapting Lemma 5 in [4], [4] in fact implies a $\sigma\sqrt{\epsilon} $ error rate. We have added the following sentence in the related work section: “In addition, [PJL20] adapted the implicit result in [4] to show that iterative thresholding algorithm achieves $O(\sigma\sqrt{\epsilon})$ error rate for sub-Gaussian covariate”
> > >
> > > **Q: (Dependence on the norm bound R) I would like to see the explicit dependence on R. Can the authors please comment on the dependence on R in the runtime, sample complexity, and the error rate for the GLMs considered in the paper? In many settings, the dependence on R turns out to be crucial, e.g., logistic regression.**
> > >
> > > A: Here we denote $R=||\beta^*||$. We tracked the dependency on $R$ in our proof and listed them as below.
> > > Given the corruption level $\epsilon$, the sample complexity here is always $O(d/\epsilon^2)$.
> > >
> > > **Gaussian regression:**
> > > Error rate: no R dependency.
> > > Iterations: poly(R) dependency.
> > > **Poisson regression:**
> > > Error rate: $\epsilon \exp(R\sqrt{\log(1/\epsilon)})$.
> > > Iterations: $\exp(R)$ dependency.
> > > **Binomial regression:**
> > > Error rate: $\tilde{O}(\epsilon \exp(R)/\sqrt{m})$ assuming that $\epsilon\le \exp(-R)$. Note that this dependency seems to be unavoidable since whenever $R = \log(1/\epsilon)$ and m = 1 (logistic regression setting) , the adversary can corrupt the labels such that $y$ comes from a halfspace in which case the best fit $\hat{\beta}$ would have infinite length, therefore no algorithm can have a non-trivial estimate for parameter recovery.
> > > Iterations: no R dependency.
> > > **General GLMs:**
> > > Error rate: $O(\epsilon \log(1/\epsilon) R)$
> > > Iterations: poly(R) dependency.
> > >
> > > **Q: (Proof sketch in the main body) It is my understanding that the proof sketch has not changed in the new version. Please let me know if that is correct. As I said in my review, the proof sketch needs to be clearer.**
> > >
> > > A: We have added the formal definition of resilience in the proof sketch, made changes in a few places (marked in blue) and fixed all the typos.

---

> > > > ### Comment · Reviewer_nLxp · 2022-08-07
> > > > **Thank you for your response**
> > > >
> > > > I thank the authors for their detailed response to my comments. Their response has answered my concerns, and I continue to recommend acceptance.
> > > >
> > > > Dependence on the norm $R$: Thank you for reporting these results. I would suggest adding a section in the Appendix with these claims; please also include a (very) brief proof sketch to derive these claimed bounds.

---

### Official Review · Reviewer_8oay · 2022-07-13

**Rating:** 7
**Confidence:** 3
**Soundness:** 3 good
**Presentation:** 3 good
**Contribution:** 3 good

**Summary:**

This is a theoretical work studying a widely used heuristic in classification with label corruption. The algorithm of interest is trimmed maximum likelihood estimation, i.e., finding the parameters with maximum likelihood over all subsets of training data that have the same size as the clean set. The algorithm is widely used but existing theoretical guarantees only apply to Gaussian regression model. The paper proved improved and minimax near-optimal results for a broader class of generalized linear models.

**Questions:**

* Should there be an asumption that eps < 1?
* In algorithm 1, |T| = (1 - 2eps) n is assigned to the new S_hat. Should it be (1-eps)n according to the description elsewhere and in [26]? This is related to my next question
* If |S_hat| = (1 - eps)n, then how could you ensure that T\S_hat has the same number of data points as S_hat \intersect E?


**Limitations:**

This is pure theoretical work. The author pointed out the limitations in applying their SDP-based algorithm to covariate corruptions problems.

**Strengths And Weaknesses:**

## Strengths
* The result is a clear step beyond existing work towards the theoretical understanding of a heuristic but widely used algorithm.
* Particularly, the results improved existing constant error bounds to ~O(eps) (the minimax optimal), with a dependence on the corruption level.
* The writing is clear. Proof sketch is appreciated.

## Weaknesses

* The proof relies on sub-gaussian data and its “resilience” property, which can be a strong assumption that the original data is from “nice” distribution.

---

> ### Author Response · Authors · 2022-08-02
> **Response to reviewer 8oay**
>
> Thanks so much for your thoughtful review and questions. Please find responses below.
>
> **Q: Should there be an assumption that $\epsilon < 1$?**
>
> A: Yes, we actually assume $\epsilon<c$ for a small constant $c$ throughout this paper and did not carefully track the value of $c$. We have explicitly added this assumption in section 3.
>
> **Q: In algorithm 1, $|T| = (1 - 2\epsilon)n$ is assigned to the new $\hat{S}$. Should it be $(1-\epsilon)n$ according to the description elsewhere and in [26]?**
>
> A: Since $S^{(0)}$ only has $(1-\epsilon)n$ examples, we need to trim another set of size $\epsilon n$ for the iterative trimming algorithm, which results in a set of size $(1-2\epsilon) n$
>
> **Q: If $|\hat{S}| = (1 -\epsilon)n$, then how could you ensure that $T\setminus\hat{S}$ has the same number of data points as $\hat{S} \cap E$?**
>
> A: If $|\hat{S}| = (1-\epsilon)n$, good data $|T| = |(1-\epsilon)n|$, bad data $|E| = \epsilon n$, we actually do have $|T\\hat{S}| = |\hat{S}\cap E|$. However, during step 3-7 we have $|\hat{S}| =  (1 - 2\epsilon)n$, good set $|T| >=(1-2\epsilon)n$, bad set $|E|\le\epsilon n$  it is not necessarily true that $|T\\hat{S}| = |\hat{S}\cap E|$. But we can redefine $E$ to be a set of size $\epsilon n$ which contains all the bad data and some good data. Under this new definition, we will again have $|T\setminus \hat{S}| = |\hat{S}\cap E|$. We will add this explanation to the proof.

---

> > ### Comment · Reviewer_8oay · 2022-08-08
> > **Thanks for the response.**
> >
> > I have read the authors' response and appreciate the clarification. I keep my original score and vote for acceptance.

---

### Meta-Review · Area_Chair_wx5P · 2022-08-27

**Recommendation:** Accept
**Confidence:** Certain

**Metareview:**

This paper analyzes Trimmed Maximum Likelihood Estimation (TML) for generalized linear model when the data is corrupted adversarially.  Under a range of link functions, minimax near-optimal risk is shown achieved, tightening the state-of-the-art results.  In addition to label corruption, extensions were made to covariate corruption with semi-definite programming.  All the reviewers, including myself, find the paper a solid contribution to the methodology and analysis, hence a clear accept.  The rebuttal provides some useful results and insights, which can be included in the final version of the paper.


**Award:**

No

---

### Decision · Program_Chairs · 2022-09-14

Accept